# Efficient Neural Network Training via Forward and Backward Propagation Sparsification

**Xiao Zhou**[*1], **Weizhong Zhang**[*1], **Zonghao Chen**[2], **Shizhe Diao**[1], **Tong Zhang**[†1]

[1] Hong Kong University of Science and Technology, [2] Tsinghua University

xzhoubi@connect.ust.hk, zhangweizhongzju@gmail.com

czh17@mails.tsinghua.edu.cn, sdiaoaa@ust.hk, tongzhang@tongzhang-ml.org

## Abstract

Sparse training is a natural idea to accelerate the training speed of deep neural networks and save the memory usage, especially since large modern neural networks are significantly over-parameterized. However, most of the existing methods cannot achieve this goal in practice because the chain rule based gradient (w.r.t. structure parameters) estimators adopted by previous methods require dense computation at least in the backward propagation step. This paper solves this problem by proposing an efficient sparse training method with completely sparse forward and backward passes. We first formulate the training process as a continuous minimization problem under global sparsity constraint. We then separate the optimization process into two steps, corresponding to weight update and structure parameter update. For the former step, we use the conventional chain rule, which can be sparse via exploiting the sparse structure. For the latter step, instead of using the chain rule based gradient estimators as in existing methods, we propose a variance reduced policy gradient estimator, which only requires two forward passes without backward propagation, thus achieving completely sparse training. We prove that the variance of our gradient estimator is bounded. Extensive experimental results on real-world datasets demonstrate that compared to previous methods, our algorithm is much more effective in accelerating the training process, up to an order of magnitude faster.

## 1 Introduction

In the last decade, deep neural networks (DNNs) [35, 11, 38] have proved their outstanding performance in various fields such as computer vision and natural language processing. However, training such large-sized networks is still very challenging, requiring huge computational power and storage. This hinders us from exploring larger networks, which are likely to have better performance. Moreover, it is a widely-recognized property that modern neural networks are significantly over-parameterized, which means that a fully trained network can always be sparsified dramatically by network pruning techniques [9, 8, 25, 46, 20] into a small sub-network with negligible degradation in accuracy. After pruning, the inference efficiency can be greatly improved. Therefore, a natural question is *can we exploit this sparsity to improve the training efficiency?*

The emerging technique called sparse network training [10] is closely related with our question, which can obtain sparse networks by training from scratch. We can divide existing methods into two categories, i.e., *parametric* and *non-parametric*, based on whether they explicitly parameterize network structures with trainable variables (termed *structure parameters*). Empirical results [24, 34, 44, 23] demonstrate that the sparse networks they obtain have comparable accuracy with those

---

[*]Equal contribution

[†]Jointly with Google Research

35th Conference on Neural Information Processing Systems (NeurIPS 2021).

obtained from network pruning. However, most of them narrowly aim at finding a sparse subnetwork instead of simultaneously sparsifying the computation of training by exploiting the sparse structure. As a consequence, it is hard for them to effectively accelerate the training process in practice on general platforms, e.g., Tensorflow [1] and Pytorch [31]. Detailed reasons are discussed below:

- Non-parametric methods find the sparse network by repeating a two-stage procedure that alternates between weight optimization and pruning [10, 6], or by adding a proper sparsity-inducing regularizer on the weights to the objective [22, 41]. The two-stage methods prune the networks in weight space and usually require retraining the obtained subnetwork from scratch every time when new weights are pruned, which makes training process even more time-consuming. Moreover, the computation of regularized methods is dense since the gradients of a zero-valued weights/filters are still nonzero.

- All the parametric approaches estimate the gradients based on chain rule. The gradient w.r.t. the structure parameters can be nonzero even when the corresponding channel/weight is pruned. Thus, to calculate the gradient via backward propagation, the error has to be propagated through all the neurons/channels. This means that the computation of backward propagation has to be dense. Concrete analysis can be found in Section 3.

We notice that some existing methods [4, 28] can achieve training speedup by careful implementation. For example, the dense to sparse algorithm [28] removes some channels if the corresponding weights are quite small for a long time. However, these methods always need to work with a large model at the beginning epochs and consume huge memory and heavy computation in the early stage. Therefore, even with such careful implementations, the speedups they can achieve are still limited.

In this paper, we propose an efficient channel-level parametric sparse neural network training method, which is comprised of **completely sparse** (See Remark 1) forward and backward propagation. We adopt channel-level sparsity since such sparsity can be efficiently implemented on the current training platforms to save the computational cost. In our method, we first parameterize the network structure by associating each filter with a binary mask modeled as an independent Bernoulli random variable, which can be continuously parameterized by the probability. Next, inspired by the recent work [47], we globally control the network size during the whole training process by controlling the sum of the Bernoulli distribution parameters. Thus, we can formulate the sparse network training problem into a constrained minimization problem on both the weights and structure parameters (i.e., the probability). The main novelty and contribution of this paper lies in our efficient training method called *completely sparse neural network training* for solving the minimization problem. Specifically, to fully exploit the sparse structure, we separate training iteration into two parts, i.e., weight update and structure parameter update. For weight update, the conventional backward propagation is used to calculate the gradient, which can be sparsified completely because the gradients of the filters with zero valued masks are also zero. For structure parameter update, we develop a new **v**ariance **r**educed **p**olicy **g**radient **e**stimator (VR-PGE). Unlike the conventional chain rule based gradient estimators (e.g., straight through[2]), VR-PGE estimates the gradient via two forward propagations, which is completely sparse because of the sparse subnetwork. Finally, extensive empirical results demonstrate that our method can significantly accelerate the training process of neural networks.

The main contributions of this paper can be summarized as follows:

- We develop an efficient sparse neural network training algorithm with the following three appealing features:
  - In our algorithm, the computation in both forward and backward propagations is completely sparse, i.e., they do not need to go through any pruned channels, making the computational complexity significantly lower than that in standard training.
  - During the whole training procedure, our algorithm works on small sub-networks with the *target sparsity* instead of follows a dense-to-sparse scheme.
  - Our algorithm can be implemented easily on widely-used platforms, e.g., Pytorch and Tensorflow, to achieve practical speedup.

- We develop a variance reduced policy gradient estimator VR-PGE specifically for sparse neural network training, and prove that its variance is bounded.

- Experimental results demonstrate that our methods can achieve significant speed-up in training sparse neural networks. This implies that our method can enable us to explore larger-sized neural networks in the future.

**Remark 1.** *We call a sparse training algorithm **completely sparse** if both its forward and backward propagation do not need to go through any pruned channels. For such algorithms, the computational cost in forward and backward propagation cost can be roughly reduced to $\rho^2 * 100\%$, with $\rho$ being the ratio of remaining unpruned channels.*

## 2 Related Work

In this section, we briefly review the studies on neural network pruning, which refers to the algorithms that prune DNNs after fully trained, and the recent works on sparse neural network training.

### 2.1 Neural Network Pruning

Network Pruning [10] is a promising technique for reducing the model size and inference time of DNNs. The key idea of existing methods [10, 8, 46, 20, 27, 13, 48, 40, 43, 32, 16] is to develop effective criteria (e.g, weight magnitude) to identify and remove the massive unimportant weights contained in networks after training. To achieve practical speedup on general devices, some of them prune networks in a structured manner, i.e., remove the weights in a certain group (e.g., filter) together, while others prune the weights individually. It has been reported in the literature [8, 25, 46, 20] that they can improve inference efficiency and reduce memory usage of DNNs by orders of magnitudes with minor loss in accuracy, which enables the deployment of DNNs on low-power devices.

We notice that although some pruning methods can be easily extended to train sparse networks, they cannot accelerate or could even slow down the training process. One reason is they are developed in the scenario that a fully trained dense network is given, and cannot work well on the models learned in the early stage of training. Another reason is after each pruning iteration, one has to fine tune or even retrain the network for lots of epoch to compensate the caused accuracy degradation.

### 2.2 Sparse Neural Network Training

The research on sparse neural network training has emerged in the recent years. Different from the pruning methods, they can find sparse networks without pre-training a dense one. Existing works can be divided into four categories based on their granularity in pruning and whether the network structures are explicitly parameterized. To the best of our knowledge, no significant training speedups achieved in practice are reported in the literature. Table 1 summarizes some representative works.

Table 1: Some representative works in sparse neural network training.

| granularity | non-parametric | parametric |
|---|---|---|
| weight-level | [5, 6, 48, 22, 18, 29, 39, 30, 4] | [42, 37, 26, 47, 18] |
| channel-level | [41, 12] | [19, 24, 44, 26, 16] |

Weight-level non-parametric methods, e.g., [6, 10, 48, 29, 30], always adopt a two-stage training procedure that alternates between weight optimization and pruning. They differ in the schedules of tuning the prune ratio over training and layers. [10] prunes the weights with the magnitude below a certain threshold and [48, 6] gradually increase the pruning rate during training. [30, 5] automatically reallocate parameters across layers during training via controlling the global sparsity.

Channel-level non-parametric methods [12, 41] are proposed to achieve a practical acceleration in inference. [41] is a structured sparse learning method, which adds a group Lasso regularization into the objective function of DNNs with each group comprised of the weights in a filter. [12] proposes a soft filter pruning method. It zeroizes instead of hard pruning the filters with small $\ell_2$ norm, after which these filters are treated the same with other filters in training. It is obvious that these methods cannot achieve significant speedup in training since they need to calculate the full gradient in backward propagation although the forward propagation could be sparsified if implemented carefully.

Parametric methods multiply each weight/channel with a binary [47, 44, 37, 42] or continuous [24, 26, 19, 18] mask, which can be either deterministic [24, 42] or stochastic [47, 44, 26, 37, 19, 18]. The mask is always parameterized via a continuous trainable variable, i.e., structure parameter. The

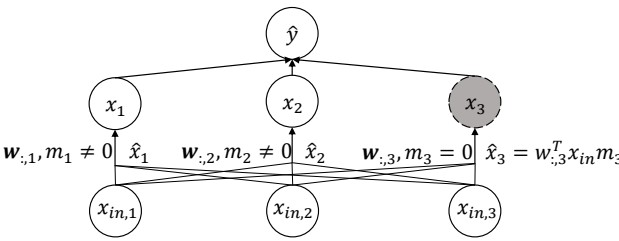

Figure 1: A fully connected network. $\boldsymbol{w}$ is the weight matrix of 1st layer, $m_i$ is the mask of $i$-th neuron; $\hat{y}$, $\hat{\boldsymbol{x}}_{in}$ and $\hat{x}_i$ are the output, input and preactivation. The 3rd neuron (in grey) is pruned.

sparsity is achieved by adding sparsity-inducing regularizers on the masks. The novelties of these methods lie in estimating the gradients w.r.t structure parameters in training. To be precise,

- *Deterministic Binary Mask.* [42] parameterizes its deterministic binary mask as a simple step function and estimates the gradients via sigmoid straight through estimator (STE) [2].

- *Deterministic Continuous Mask.* [24] uses the linear coefficients of batch normalization (BN) as a continuous mask and enforces most of them to 0 by penalizing the objective with $\ell_1$ norm of the coefficients. [18] defines the mask as a soft threshold function with learnable threshold. These methods can estimate the gradients via standard backward propagation.

- *Stochastic Binary Mask.* [44, 37] model the mask as a bernoulli random variable and the gradients w.r.t. the parameters of bernoulli distributions are estimated via STE. [47] estimates the gradients via Gumbel-Softmax trick [15], which is more accurate than STE.

- *Stochastic Continuous Mask.* [26, 19] parameterize the mask as a continuous function $g(c, \epsilon)$, which is differentiable w.r.t. $c$, and $\epsilon$ is a parameter free noise, e.g., Gaussian noise $\mathcal{N}(0, 1)$. In this way, the gradients can be calcuated via conventional backward propagation.

Therefore, we can see that all of these parametric methods estimate the gradients of the structure parameters based on the chain rule in backward propagation. This makes the training iteration cannot be sparsified by exploiting the sparse network structure. For the details, please refer to Section 3.

## 3  Why Existing Parameteric Methods Cannot Achieve Practical Speedup?

In this section, we reformulate existing parametric channel-level methods into a unified framework to explain why they cannot accelerate the training process in practice.

Notice that convolutional layer can be viewed as a generalized fully connected layer, i.e., viewing the channels as neurons and convolution of two matrices as a generalized multiplication (see [7]). Hence, for simplicity, we consider the fully connected network in Figure 1. Moreover, since the channels in CNNs are corresponding to the neurons in fully connected networks, we consider *neuron-level instead of weight-level* sparse training in our example.

As discussed in Section 2, existing methods parameterize the 4 kinds of mask in the following ways:

(i): $m_i = \phi(s_i)$;  (ii): $m_i = \psi(s_i)$;  (iii): $m_i = g(s_i, \epsilon), \epsilon \sim \mathcal{N}(0, 1)$;  (iv): $m \sim \text{Bern}(p_i(s))$,

where the function $\phi(s_i)$ is binary, e.g., step function; $\psi(s_i)$ is a continuous function; $g(s_i, \epsilon)$ is differentiable w.r.t. $s_i$. All existing methods estimate the gradient of the loss $\ell(\hat{y}, y)$ w.r.t. $s_i$ based on chain rule, which can be formulated into a unified form below.

Specifically, we take the pruned neuron $x_3$ in Figure 1 as an example, the gradient is calculated as

$$\nabla_{s_3} \ell(\hat{y}, y) = \underbrace{\frac{\partial \ell(\hat{y}, y)}{\partial \hat{x}_3}}_{a} \underbrace{\left( \boldsymbol{w}_{:,3}^{\top} \boldsymbol{x}_{in} \right)}_{forward} \frac{\partial m_3}{\partial s_3}. \tag{1}$$

Existing parametric methods developed different ways to estimate $\frac{\partial m_3}{\partial s_3}$. Actually, for cases (ii) and (iii), the gradients are well-defined and thus can be calculated directly. STE is used to estimate the gradient in case (i) [42]. For cases (iv), [44, 37, 47] adopt STE and Gumbel-Softmax.

In Eqn.(1), the term (a) is always nonzero especially when $\hat{x}_3$ is followed by BN. Hence, we can see that even for the pruned neuron $x_3$, the gradient $\frac{\partial m_3}{\partial s_3}$ can be nonzero in all four cases. This means the backward propagation has to go though all the neurons/channels, leading to dense computation.

At last, we can know from Eqn.(1) that forward propagation in existing methods cannot be completely sparse. Although $\boldsymbol{w}_{:,3}^\top \boldsymbol{x}_{in}$ can be computed sparsely as in general models $\boldsymbol{x}_{in}$ could be a sparse tensor of a layer with some channels being pruned, we need to calculate it for *each* neuron via forward propagation to calculate RHS of Eqn.(1). Thus, even if carefully implemented, the computational cost of forward propagation can only be reduced to $\rho * 100\%$ instead of $\rho^2 * 100\%$ as in inference.

That's why we argue that existing methods need dense computation at least in backward propagation. So they cannot speed up the training process effectively in practice.

**Remark 2.** *The authors of GrowEfficient [44] confirmed that actually they also calculated the gradient of $q_c$ w.r.t. $s_c$ in their Eqn.(6) via STE even if $q_c = 0$. Thus need dense backward propagation.*

## 4 Channel-level Completely Sparse Neural Network Training

Below, we present our sparse neural network training framework and the efficient training algorithm.

### 4.1 Framework of Channel-level Sparse Training

Given a convolutional network $f(x; \boldsymbol{w})$, let $\{\mathcal{F}_c : c \in \mathcal{C}\}$ be the set of filters with $\mathcal{C}$ being the set of indices of all the channels. To parameterize the network structure, we associate each $\mathcal{F}_c$ with a binary mask $m_c$, which is an independent Bernoulli random variable. Thus, each channel is computed as

$$\boldsymbol{x}_{\text{out, c}} = \boldsymbol{x}_{in} * (\mathcal{F}_c m_c),$$

with $*$ being the convolution operation. Inspired by [47], to avoid the problems, e.g., gradient vanishing, we parameterize $m_c$ directly on the probability $s_c$, i.e., $m_c$ equals to 1 and 0 with the probabilities $s_c$ and $1 - s_c$, respectively. Thus, we can control the channel size by the sum of $s_c$. Following [47], we can formulate channel-level sparse network training into the following framework:

$$\min_{\boldsymbol{w}, \boldsymbol{s}} \; \mathbb{E}_{p(\boldsymbol{m}|\boldsymbol{s})} \, \mathcal{L}(\boldsymbol{w}, \boldsymbol{m}) := \frac{1}{N} \sum_{i=1}^{N} \ell \left( f\left(\mathbf{x}_i; \boldsymbol{w}, \boldsymbol{m}\right), \mathbf{y}_i \right) \tag{2}$$

$$s.t. \; \boldsymbol{w} \in \mathbb{R}^n, \boldsymbol{s} \in \mathcal{S} := \{ \boldsymbol{s} \in [0, 1]^{|\mathcal{C}|} : \mathbf{1}^\top \boldsymbol{s} \leq K \},$$

where $\{(\mathbf{x}_i, \mathbf{y}_i)\}_{i=1}^{N}$ is the training dataset, $\boldsymbol{w}$ is the weights of the original network, $f(\cdot; \cdot, \cdot)$ is the pruned network, and $\ell(\cdot, \cdot)$ is the loss function, e.g, cross entropy loss. $K = \rho|\mathcal{C}|$ controls the remaining channel size with $\rho$ being the remaining ratio of the channels.

**Discussion.** We'd like to point out that although our framework is inspired by [47], our main contribution is the efficient solver comprised of completely sparse forward/backward propagation for Problem (2). Moreover, our framework can prune the weights in fully connected layers together, since we can associate each weight with an independent mask.

### 4.2 Completely Sparse Training with Variance Reduced Policy Gradient

Now we present our completely sparse training method, which can solve Problem (2) via *completely* sparse forward and backward propagation. The key idea is to separate the training iteration into filter update and structure parameter update so that the sparsity can be fully exploited.

#### 4.2.1 Filter Update via Completely Sparse Computation

It is easy to see that the computation of the gradient w.r.t. the filters can be sparsified completely. To prove this point, we just need to clarify the following two things:

- *We do not need to update the filters corresponding to the pruned channels.* Consider a pruned channel $c$, i.e., $m_c = 0$, then due to the chain rule, we can have

$$\frac{\partial \ell \left( f\left(\mathbf{x}_i; \boldsymbol{w}, \boldsymbol{m}\right) \right)}{\partial \mathcal{F}_c} = \frac{\partial \ell \left( f\left(\mathbf{x}_i; \boldsymbol{w}, \boldsymbol{m}\right) \right)}{\partial \boldsymbol{x}_{out,c}} \frac{\partial \boldsymbol{x}_{out,c}}{\partial \mathcal{F}_c} \equiv 0,$$

the last equation holds since $\boldsymbol{x}_{out,c} \equiv 0$. This indicates that the gradient w.r.t the pruned filter $\mathcal{F}_c$ is always 0, and thus $\mathcal{F}_c$ does not need to be updated.

- *The error cannot pass the pruned channels via backward propagation.* Consider a pruned channel $c$, we denote its output before masking as $\hat{\boldsymbol{x}}_{out,c} = \boldsymbol{x}_{in} * \mathcal{F}_c$, then the error propagating through this channel can be computed as

$$\frac{\partial \ell \left( f \left( \mathbf{x}_i; \boldsymbol{w}, \boldsymbol{m} \right) \right)}{\partial \hat{\boldsymbol{x}}_{out,c}} = \frac{\partial \ell \left( f \left( \mathbf{x}_i; \boldsymbol{w}, \boldsymbol{m} \right) \right)}{\partial \boldsymbol{x}_{out,c}} \frac{\partial \boldsymbol{x}_{out,c}}{\hat{\boldsymbol{x}}_{out,c}} \equiv 0.$$

This demonstrates that to calculate the gradient w.r.t. the unpruned filters, the backward propagation does not need to go through any pruned channels.

Therefore, the filters can be updated via completely sparse backward propagation.

### 4.2.2 Structure Parameter Update via Variance Reduced Policy Gradient

We notice that **p**olicy **g**radient **e**stimator (PGE) can estimate the gradient via forward propagation, avoiding the pathology of chain rule based estimators as dicussed in Section 3. For abbreviation, we denote $\mathcal{L}(\boldsymbol{w}, \boldsymbol{m})$ as $\mathcal{L}(\boldsymbol{m})$ since $\boldsymbol{w}$ can be viewed as a constant here. The objective can be written as

$$\Phi(\boldsymbol{s}) = \mathbb{E}_{p(\boldsymbol{m}|\boldsymbol{s})} \mathcal{L}(\boldsymbol{m}),$$

which can be optimized using gradient descent:

$$\boldsymbol{s} \leftarrow \boldsymbol{s} - \eta \nabla \Phi(\boldsymbol{s}).$$

with learning rate $\eta$. One can obtain a stochastic unbiased estimate of the gradient $\nabla \Phi(\boldsymbol{s})$ using PGE:

$$\nabla \Phi(\boldsymbol{s}) = \mathbb{E}_{p(\boldsymbol{m}|\boldsymbol{s})} \mathcal{L}(\boldsymbol{m}) \nabla_{\boldsymbol{s}} \ln p(\boldsymbol{m}|\boldsymbol{s}), \tag{PGE}$$

leading to Policy Gradient method, which may be regarded as a stochastic gradient descent algorithm:

$$\boldsymbol{s} \leftarrow \boldsymbol{s} - \eta \mathcal{L}(\boldsymbol{m}) \nabla_{\boldsymbol{s}} \ln p(\boldsymbol{m}|\boldsymbol{s}). \tag{3}$$

In Eqn.(3), $\mathcal{L}(\boldsymbol{m})$ can be computed via completely sparse forward propagation and the computational cost of $\nabla_{\boldsymbol{s}} \ln p(\boldsymbol{m}|\boldsymbol{s}) = \frac{\boldsymbol{m}-\boldsymbol{s}}{\boldsymbol{s}(1-\boldsymbol{s})}$ is negligible, therefore PGE is computationally efficient.

However, in accordance with the empirical results reported in [33, 15], we found that standard PGE suffers from high variance and does not work in practice. Below we will develop a **V**ariance **R**educed **P**olicy **G**radient **E**stimator (VR-PGE) starting from theoretically analyzing the variance of PGE.

Firstly, we know that this variance of PGE is

$$\mathbb{E}_{p(\boldsymbol{m}|\boldsymbol{s})} \mathcal{L}^2(\boldsymbol{m}) \|\nabla_{\boldsymbol{s}} \ln p(\boldsymbol{m}|\boldsymbol{s})\|_2^2 - \|\nabla \Phi(\boldsymbol{s})\|_2^2,$$

which can be large because $\mathcal{L}(\boldsymbol{m})$ is large.

Mean Field theory [36] indicates that, while $\mathcal{L}(\boldsymbol{m})$ can be large, the term $\mathcal{L}(\boldsymbol{m}) - \mathcal{L}(\boldsymbol{m}')$ is small when $\boldsymbol{m}$ and $\boldsymbol{m}'$ are two independent masks sampled from a same distribution $p(\boldsymbol{m}|\boldsymbol{s})$ (see the appendix for the details). This means that we may consider the following variance reduced preconditioned policy gradient estimator:

$$\mathbb{E}_{\boldsymbol{m}' \sim p(\boldsymbol{m}'|\boldsymbol{s})} \mathbb{E}_{\boldsymbol{m} \sim p(\boldsymbol{m}|\boldsymbol{s})} \left( \mathcal{L}(\boldsymbol{m}) - \mathcal{L}(\boldsymbol{m}') \right) H^\alpha(\boldsymbol{s}) \nabla_{\boldsymbol{s}} \ln p(\boldsymbol{m}|\boldsymbol{s}), \tag{VR-PGE}$$

where $H^\alpha(\boldsymbol{s})$ is a specific diagonal preconditioning matrix

$$H^\alpha(\boldsymbol{s}) = \text{diag} \left( \boldsymbol{s} \circ (1 - \boldsymbol{s}) \right)^\alpha, \tag{4}$$

with $\alpha \in (0, 1)$ and $\circ$ being the element-wise product. It plays a role as adaptive step size and it is shown that this term can reduce the variance of the stochastic PGE term $\nabla_{\boldsymbol{s}} \ln p(\boldsymbol{m}|\boldsymbol{s})$. The details can be found in the appendix. Thus $\Phi(\boldsymbol{s})$ can be optimized via:

$$\boldsymbol{s} \leftarrow \boldsymbol{s} - \eta \left( \mathcal{L}(\boldsymbol{m}) - \mathcal{L}(\boldsymbol{m}') \right) H^\alpha(\boldsymbol{s}) \nabla_{\boldsymbol{s}} \ln p(\boldsymbol{m}|\boldsymbol{s}). \tag{5}$$

In our experiments, we set $\alpha$ to be $\frac{1}{2}$ for our estimator VR-PGE. The theorem below demonstrates that VR-PGE can have bounded variance.

**Algorithm 1** Completely Sparse Neural Network Training

---
**Input:** target remaining ratio $\rho$, a dense network $\boldsymbol{w}$, the step size $\eta$, and parameter $\alpha$ in (4) .

1: Initialize $\boldsymbol{w}$, let $\boldsymbol{s} = \rho\mathbf{1}$.
2: **for** training epoch $t = 1, 2 \ldots T$ **do**
3:     **for** each training iteration **do**
4:         Sample mini batch of data $\mathcal{B} = \{(\mathbf{x}_1, \mathbf{y}_1), \ldots, (\mathbf{x}_B, \mathbf{y}_B)\}$.
5:         Sample $\boldsymbol{m}^{(i)}$ from $p(\boldsymbol{m}|\boldsymbol{s})$, $i = 1, 2$.
6:         Update $\boldsymbol{s}$ and $\boldsymbol{w}$
        $\boldsymbol{s} \leftarrow \text{proj}_{\mathcal{S}}(\boldsymbol{z})$ with $\boldsymbol{z} = \boldsymbol{s} - \eta \left(\mathcal{L}_{\mathcal{B}}(\boldsymbol{w}, \boldsymbol{m}^{(1)}) - \mathcal{L}_{\mathcal{B}}(\boldsymbol{w}, \boldsymbol{m}^{(2)})\right) H^{\alpha}(\boldsymbol{s})\frac{\boldsymbol{m}^{(1)}-\boldsymbol{s}}{\boldsymbol{s}(1-\boldsymbol{s})}$,
        $\boldsymbol{w} \leftarrow \boldsymbol{w} - \eta\nabla_{\boldsymbol{w}}\mathcal{L}_{\mathcal{B}}\left(\boldsymbol{w}, \boldsymbol{m}^{(1)}\right)$
7:     **end for**
8: **end for**
9: **return** A pruned network $\boldsymbol{w} \circ \boldsymbol{m}$ by sampling a mask $\boldsymbol{m}$ from the distribution $p(\boldsymbol{m}|\boldsymbol{s})$.

---

**Theorem 1.** *Suppose $\boldsymbol{m}$ and $\boldsymbol{m}'$ are two independent masks sampled from the Bernoulli distribution $p(\boldsymbol{m}|\boldsymbol{s})$, then for any $\alpha \in [\frac{1}{2}, 1)$ and $\boldsymbol{s} \in (0, 1)^{|\mathcal{C}|}$, the variance is bounded for*

$$\left(\mathcal{L}(\boldsymbol{m}) - \mathcal{L}(\boldsymbol{m}')\right) H^{\alpha}(\boldsymbol{s})\nabla_{\boldsymbol{s}} \ln p(\boldsymbol{m}|\boldsymbol{s})$$

Finally, we provide a complete view of our sparse training algorithm in Algorithm 1, which is essentially a projected stochastic gradient descent equipped with our efficient gradient estimators above. The projection operator in Algorithm 1 can be computed efficiently using Theorem 1 of [47].

**Discussion.** In our algorithm, benefited from our constraint on $\boldsymbol{s}$, the channel size of the neural network during training can be strictly controlled. This is in contrast with GrowEfficient [44], which ultilizes regularizer term to control the model size and has situations where model size largely drift away from desired. This will have larger demand for the GPU memory storage and have more risk that memory usage may explode, especially when we utilize sparse learning to explore larger models. Moreover, our forward and backward propagations are completely sparse, i.e., they do not need to go through any pruned channels. Therefore, the computational cost of each training iteration can be roughly reduced to $\rho^2 * 100\%$ of the dense network.

## 5 Experiments

In this section, we conduct a series of experiments to demonstrate the outstanding performance of our method. We divide the experiments into five parts. In part one, we compare our method with several state-of-the-art methods on CIFAR-10 [17] using VGG-16 [35], ResNet-20 [11] and WideResNet-28-10 [45] to directly showcase the superiority of our method. In part two, we directly compare with state-of-the-art method GrowEfficient [44] especially on extremely sparse regions, and on two high capacity networks VGG-19 [35] and ResNet-32 [11] on CIFAR-10/100 [17]. In part three, we conduct experiments on a large-scale dataset ImageNet [3] with ResNet-50 [11] and MobileNetV1 [14] and compare with GrowEfficient [44] across a wide sparsity region. In part four, we present the train-computational time as a supplementary to the conceptual train-cost savings to justify the applicability of sparse training method into practice. In part five, we present further analysis on epoch-wise train-cost dynamics and experimental justification of variance reduction of VR-PGE. Due to the space limitation, we postpone the experimental configurations, calculation schemes on train-cost savings and train-computational time and additional experiments into appendix.

### 5.1 VGG-16, ResNet-20 and WideResNet-28-10 on CIFAR-10

Table 2 presents Top-1 validation accuracy, parameters, FLOPs and train-cost savings comparisons with channel pruning methods L1-Pruning [20], SoftNet [12], ThiNet [27], Provable [21] and sparse training method GrowEfficient [44]. SoftNet can train from scratch but requires completely dense computation. Other pruning methods all require pretraining of dense model and multiple rounds of pruning and finetuning, which makes them slower than vanilla dense model training. Therefore the train-cost savings of these methods are below $1\times$ and thus shown as ("-") in Table 2.

Table 2: Comparison with the channel pruning methods L1-Pruning [20], SoftNet [12], ThiNet [27], Provable [21] and one channel sparse training method GrowEfficient [44] on CIFAR-10.

| Model | Method | Val Acc(%) | Params(%) | FLOPs(%) | Train-Cost Savings(×) |
|-------|--------|-----------|-----------|----------|----------------------|
| VGG-16 | Original | 92.9 | 100 | 100 | 1× |
|  | L1-Pruning | 91.8 | 19.9 | 19.9 | - |
|  | SoftNet | 92.1 | 36.0 | 36.1 | - |
|  | ThiNet | 90.8 | 36.0 | 36.1 | - |
|  | Provable | 92.4 | 5.7 | 15.0 | - |
|  | GrowEfficient | 92.5 | 5.0 | 13.6 | 1.22× |
|  | Ours | **92.5** | **4.4** | **8.7** | **8.69×** |
| ResNet-20 | Original | 91.3 | 100 | 100 | 1× |
|  | L1-Pruning | 90.9 | 55.6 | 55.4 | - |
|  | SoftNet | 90.8 | 53.6 | 50.6 | - |
|  | ThiNet | 89.2 | 67.1 | 67.3 | - |
|  | Provable | 90.8 | 37.3 | 54.5 | - |
|  | GrowEfficient | 90.91 | 35.8 | 50.2 | 1.13× |
|  | Ours | **90.93** | **35.1** | **36.1** | **2.09×** |
| WRN-28-10 | Original | 96.2 | 100 | 100 | 1× |
|  | L1-Pruning | 95.2 | 20.8 | 49.5 | - |
|  | GrowEfficient | 95.3 | 9.3 | 28.3 | 1.17× |
|  | Ours | **95.6** | **8.4** | **7.9** | **9.39×** |

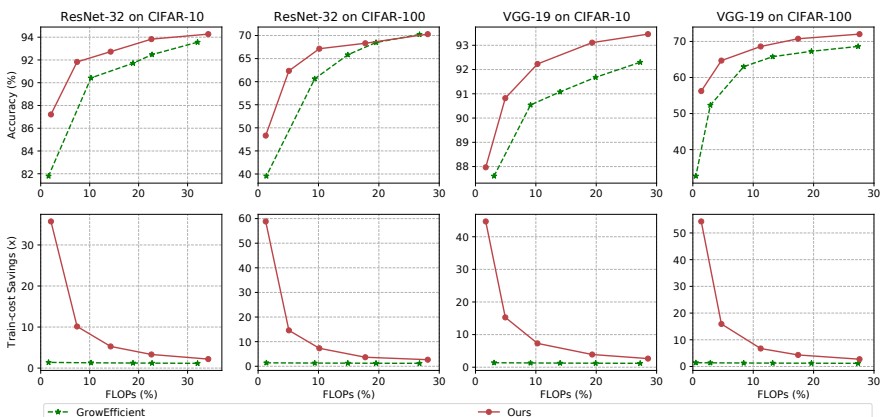

Figure 2: Comparison of Top-1 Validation Accuracy and Train-cost Savings on CIFAR-10/100.

GrowEfficient [44] is a recently proposed state-of-the-art channel-level sparse training method showing train-cost savings compared with dense training. As described in Section 3, GrowEfficient features completely dense backward and partially sparse forward pass, making its train-cost saving limited by $\frac{3}{2}$. By contrast, the train-cost savings of our method is not limited by any constraint. The details of how train-cost savings are computed can be found in appendix.

Table 2 shows that our method generally exhibits better performance in terms of validation accuracy, parameters and particularly FLOPs. In terms of train-cost savings, our method shows at least 1.85× speed-up against GrowEfficient [44] and up to 9.39× speed-up against dense training.

## 5.2 Wider Range of Sparsity on CIFAR-10/100 on VGG-19 and ResNet-32

In this section, we explore sparser regions of training efficiency to present a broader comparision with state-of-the-art channel sparse training method GrowEfficient [44].

We plot eight figures demonstrating the relationships between the Top-1 validation accuracy, FLOPs and train-cost savings. We find that our method generally achieves higher accuracy under same FLOPs

Table 3: Comparison with the channel pruning methods L1-Pruning [20], SoftNet [12], Provable [21] and one channel sparse training method GrowEfficient [44] on ImageNet-1K.

| Model | Method | Val Acc(%) | Params(%) | FLOPs(%) | Train-Cost Savings($\times$) |
|---|---|---|---|---|---|
| | Original | 77.0 | 100 | 100 | $1\times$ |
| | L1-Pruning | 74.7 | 85.2 | 77.5 | - |
| | SoftNet | 74.6 | - | 58.2 | - |
| ResNet-50 | Provable | 75.2 | 65.9 | 70.0 | - |
| | GrowEfficient | 75.2 | 61.2 | 50.3 | $1.10\times$ |
| | Ours | **76.0** | 48.2 | 46.8 | $1.60\times$ |
| | Ours | 73.5 | 27.0 | 24.7 | $3.02\times$ |
| | Ours | 69.3 | **10.8** | **10.1** | **7.36**$\times$ |

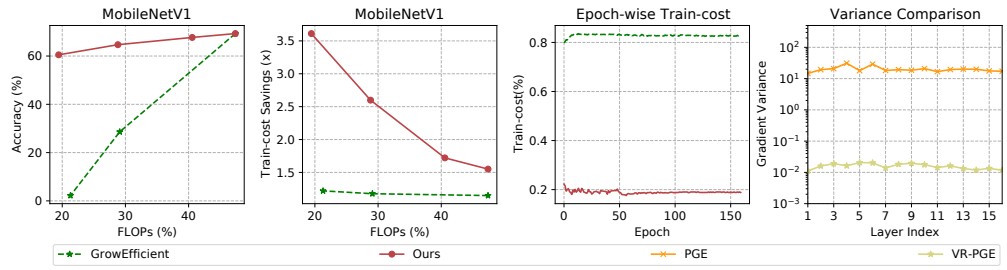

Figure 3: Top-1 Validation Accuracy and Train-cost Savings on MobileNetV1 on ImageNet. Epoch-wise Train-cost and Variance Comparison on VGG-19 on CIFAR-10.

settings. To be noted, the train-cost savings of our method is drastically higher than GrowEfficient [44], reaching up to $58.8\times$ when sparisty approches 1.56% on ResNet-32 on CIFAR-100, while the speed-up of GrowEfficient is limited by $\frac{3}{2}$.

### 5.3 ResNet-50 and MobileNetV1 on ImageNet-1K

In this section, we present the performance boost obtained by our method on ResNet-50 and Mo-bileNetV1 on ImageNet-1K [3]. Our method searches a model with 76.0% Top-1 accuracy, 48.2% parameters and 46.8% FLOPs beating all compared state-of-the-art methods. The train-cost saving comes up to $1.60\times$ and is not prominent due to the accuracy constraint to match up with compared methods. Therefore we give a harder limit to the channel size and present sparser results on the same Table 3, reaching up to $7.36\times$ speed-up while still preserving 69.3% Top-1 accuracy. For the already compact model MobileNetV1, we plot two figures in Figure 3 comparing with GrowEfficient [44]. We find that our method is much stabler in sparse regions and obtains much higher train-cost savings.

### 5.4 Actual Training Computational Time Testing

In this section, we provide actual training computational time on VGG-19 and CIFAR-10. The GPU in test is RTX 2080 Ti and the deep learning framework is Pytorch [31]. The intent of this section is to justify the feasibility of our method in reducing actual computational time cost, rather than staying in conceptual training FLOPs reduction. The computational time cost is measured by wall clock time, focusing on forward and backward propagation. We present training computational time in Table 4 with varying sparsity as in Figure 2. It shows that the computational time savings increases steadily with the sparisty. We also notice the gap between the savings in FLOPS and computational time. The gap comes from the difference between FLOPs and actual forward/backward time. More specifically, forward/backward time is slowed down by data-loading processes and generally affected by hardware latency and throughput, network architecture, etc. At extremely sparse regions, the pure computational time of sparse networks only occupies little of the forward/backward time and the cost of data management and hardware latency dominates the wall-clock time. Despite this gap, it can be expected that our train-cost savings can be better translated into real speed-up in exploring large

models where the pure computational time dominates the forward/backward time, which promises a bright future for making training infeasibly large models into practice.

Table 4: Train-computational Time on VGG-19 with CIFAR-10. The computational time saving is not as prominent as train-cost savings while still achieving nearly an order of reduction, preserving 87.97% accuracy.

| Model | Val Acc(%) | Params(%) | FLOPs(%) | Train-Cost Savings($\times$) | Train-Computational Time(min) |
|---|---|---|---|---|---|
| | 93.84 | 100.00 | 100.00 | $1.00\times$ | 21.85 ($1.00\times$) |
| | 93.46 | 23.71 | 28.57 | $2.64\times$ | 14.04 ($1.55\times$) |
| VGG-19 | 93.11 | 12.75 | 19.33 | $3.89\times$ | 10.43 ($2.09\times$) |
| | 92.23 | 6.69 | 10.27 | $7.30\times$ | 6.83 ($3.20\times$) |
| | 90.82 | 3.06 | 4.94 | $15.28\times$ | 4.86 ($4.50\times$) |
| | 87.97 | 0.80 | 1.70 | $44.68\times$ | 2.95 ($7.41\times$) |

## 5.5 Further Analysis

**[Epoch-wise Train-cost Dynamics of Sparse Training Process]** We plot the train-cost dynamics in Figure 3. The vertical label is the ratio of train-cost to dense training, the inverse of train-cost savings. This demonstrates huge difference between our method and GrowEfficient [44]. The model searched by our method exhibits 92.73% Top-1 accuracy, 16.68% parameters, 14.28% FLOPs with $5.28\times$ train-cost savings, while the model searched by GrowEfficient exhibits 92.47% Top-1 accuracy, 18.08% parameters, 22.74% FLOPs with $1.21\times$ train-cost savings.

**[Experimental Verification of Variance Reduction of VR-PGE against PGE]** We plot the mean of variance of gradients of channels from different layers. The model checkpoint and input data are selected randomly. The gradients are calculated in two approaches, VR-PGE and PGE. From the rightmost graph of Figure 3, we find that the VR-PGE reduces variance significantly, up to 3 orders of magnitude.

## 6 Conclusion

This paper proposes an efficient sparse neural network training method with completely sparse forward and backward passes. A novel gradient estimator named VR-PGE is developed for updating structure parameters, which estimates the gradient via two sparse forward propagation. We theoretically proved that VR-PGE has bounded variance. In this way, we can separate the weight and structure update in training and making the whole training process completely sparse. Emprical results demonstrate that the proposed method can significantly accelerate the training process of DNNs in practice. This enables us to explore larger-sized neural networks in the future.

## Acknowledgments and Disclosure of Funding

This work is supported by GRF 16201320.

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
