# Supplemental Material: Efficient Neural Network Training via Forward and Backward Propagation Sparsification

This appendix can be divided into four parts. To be precise,

1. Section A gives the detailed proof of Theorem 1 and discuss the convergence of our method.

2. Section B present experimental configurations of this work.

3. Section C present calculation schemes on train-cost savings and train-computational time.

4. Section D discusses the potentials and limitations of this work.

## A    Proof of Theorem 1

### A.1    Properties of Overparameterized Deep Neural Networks

Before giving the detailed proof, we would like to present the following two properties of overparameterized deep neural networks, which are implied by the latest studies based on the mean field theory. We will empirically verify these properties in this section and adopt them as assumptions in our proof.

**Property 1.** *Given the probability $\boldsymbol{s}$ and the weights $\boldsymbol{w}$ for an overparameterized deep neural network, then for two independent masks $\boldsymbol{m}$ and $\boldsymbol{m}'$ sampled from $p(\cdot|\boldsymbol{s})$, $\mathcal{L}(\boldsymbol{m}) - \mathcal{L}(\boldsymbol{m}')$ is always small. That is*

$$V(\boldsymbol{s}) := \mathbb{E}_{\boldsymbol{m}\sim p(\cdot|\boldsymbol{s})}\mathbb{E}_{\boldsymbol{m}'\sim p(\cdot|\boldsymbol{s})}\left(\mathcal{L}(\boldsymbol{m}) - \mathcal{L}(\boldsymbol{m}')\right)^2 \tag{6}$$

*is small.*

The mean field theory based studies [39, 7] proved that discrete deep neural networks can be viewed as sampling neurons/channels from continuous networks according to certain distributions. As the numbers of neurons/channels increase, the output of discrete networks would converge to that of the continuous networks (see Theorem 3 in [39] and Theorem 1 in [7]). Although in standard neural networks we do not have the scaling operator as [39, 7] for computing the expectation, due to the batch normalization layer, the affect caused by this difference can largely be eliminated. The subnetworks $\boldsymbol{m}$ and $\boldsymbol{m}'$ here can be roughly viewed as sampled from a common continuous network. Therefore, $\mathcal{L}(\boldsymbol{m}) - \mathcal{L}(\boldsymbol{m}')$ would be always small. That's why Property 1 holds.

**Property 2.** *Given the probability $\boldsymbol{s}$ and the weights $\boldsymbol{w}$ for an overparameterized deep neural network, consider a mask $\boldsymbol{m}$ sampled from $p(\cdot|\boldsymbol{s})$, if we flip one component of $\boldsymbol{m}$, then the network would not change too much. Combined with Property 1, this can be stated as: for any $j \in \mathcal{C}$, we denote $\boldsymbol{m}_{-j}$ and $\boldsymbol{s}_{-j}$ to be all the components of $\boldsymbol{m}$ and $\boldsymbol{s}$ except the $j$-th component, and define*

$$V_{\max}(\boldsymbol{s}) := \max_{\boldsymbol{m}_j\in\{0,1\},j\in\mathcal{C}} \mathbb{E}_{\boldsymbol{m}_{-j}\sim p(\cdot|\boldsymbol{s}_{-j})}\mathbb{E}_{\boldsymbol{m}'\sim p(\cdot|\boldsymbol{s})}\left(\mathcal{L}(\boldsymbol{m}) - \mathcal{L}(\boldsymbol{m}')\right)^2,$$

*then*

$$V_{\max}(\boldsymbol{s}) \approx V(\boldsymbol{s}). \tag{7}$$

In the mean field based studies [39, 7], they model output of a neuron/channel as a expectation of weighted sum of the neurons/channels in the previous layer w.r.t. a certain distribution. Therefore, the affect of flipping one component of the mask on expectation is negligible. Therefore Property 2 holds.

## A.2 Detailed Proof

*Proof.* In this proof, we denote
$$(\mathcal{L}(\boldsymbol{m}) - \mathcal{L}(\boldsymbol{m}')) H^\alpha(\boldsymbol{s}) \nabla_{\boldsymbol{s}} \ln p(\boldsymbol{m}|\boldsymbol{s})$$
as $\mathcal{G}^\alpha(\boldsymbol{m}, \boldsymbol{m}'|\boldsymbol{s})$. Note that the total variance
$$\mathrm{Var}(\mathcal{G}^\alpha(\boldsymbol{m}, \boldsymbol{m}'|\boldsymbol{s}))$$
$$= \mathbb{E}_{\boldsymbol{m}\sim p(\cdot|\boldsymbol{s})} \mathbb{E}_{\boldsymbol{m}'\sim p(\cdot|\boldsymbol{s})} \|\mathcal{G}^\alpha(\boldsymbol{m}, \boldsymbol{m}'|\boldsymbol{s})\|_2^2 - \|\mathbb{E}_{\boldsymbol{m}\sim p(\cdot|\boldsymbol{s})} \mathbb{E}_{\boldsymbol{m}'\sim p(\cdot|\boldsymbol{s})} \mathcal{G}^\alpha(\boldsymbol{m}, \boldsymbol{m}'|\boldsymbol{s})\|_2^2,$$
we only need to prove that the term $\mathbb{E}_{\boldsymbol{m}\sim p(\cdot|\boldsymbol{s})} \mathbb{E}_{\boldsymbol{m}'\sim p(\cdot|\boldsymbol{s})} \|\mathcal{G}^\alpha(\boldsymbol{m}, \boldsymbol{m}'|\boldsymbol{s})\|_2^2$ is bounded.

We let $\boldsymbol{m}_{-j}$ and $\boldsymbol{s}_{-j}$ be all the components of $\boldsymbol{m}$ and $\boldsymbol{s}$ except the $j$-th component with $j \in \mathcal{C}$. We consider the $j$-th component of $\mathcal{G}^\alpha(\boldsymbol{m}, \boldsymbol{m}'|\boldsymbol{s})$, i.e., $\mathcal{G}_j^\alpha(\boldsymbol{m}, \boldsymbol{m}'|\boldsymbol{s})$, then $\mathbb{E}_{\boldsymbol{m}\sim p(\cdot|\boldsymbol{s})} \mathbb{E}_{\boldsymbol{m}'\sim p(\cdot|\boldsymbol{s})} \|\mathcal{G}_j^\alpha(\boldsymbol{m}, \boldsymbol{m}'|\boldsymbol{s})\|_2^2$ can be estimated as

$$\mathbb{E}_{\boldsymbol{m}\sim p(\cdot|\boldsymbol{s})} \mathbb{E}_{\boldsymbol{m}'\sim p(\cdot|\boldsymbol{s})} \left(\mathcal{G}_j^\alpha(\boldsymbol{m}, \boldsymbol{m}'|\boldsymbol{s})\right)^2$$
$$= \mathbb{E}_{\boldsymbol{m}\sim p(\cdot|\boldsymbol{s})} \mathbb{E}_{\boldsymbol{m}'\sim p(\cdot|\boldsymbol{s})} \left(\mathcal{L}(\boldsymbol{m}) - \mathcal{L}(\boldsymbol{m}')\right)^2 [H^\alpha(\boldsymbol{s}) \nabla_{\boldsymbol{s}} \ln p(\boldsymbol{m}|\boldsymbol{s})]_j^2$$
$$= \mathbb{E}_{\boldsymbol{m}\sim p(\cdot|\boldsymbol{s})} \mathbb{E}_{\boldsymbol{m}'\sim p(\cdot|\boldsymbol{s})} \left(\mathcal{L}(\boldsymbol{m}) - \mathcal{L}(\boldsymbol{m}')\right)^2 \left(\boldsymbol{s}_j^{2\alpha}(1 - \boldsymbol{s}_j)^{2\alpha} \frac{(\boldsymbol{m}_j - \boldsymbol{s}_j)^2}{\boldsymbol{s}_j^2(1 - \boldsymbol{s}_j)^2}\right) \tag{8}$$
$$= \mathbb{E}_{\boldsymbol{m}\sim p(\cdot|\boldsymbol{s})} \mathbb{E}_{\boldsymbol{m}'\sim p(\cdot|\boldsymbol{s})} \left(\mathcal{L}(\boldsymbol{m}) - \mathcal{L}(\boldsymbol{m}')\right)^2 \left(\boldsymbol{s}_j^{2(\alpha-1)}(1 - \boldsymbol{s}_j)^{2(\alpha-1)}(\boldsymbol{m}_j - \boldsymbol{s}_j)^2\right)$$
$$= \mathbb{E}_{\boldsymbol{m}_j\sim p(\cdot|\boldsymbol{s}_j)} \left(\mathbb{E}_{\boldsymbol{m}_{-j}\sim p(\cdot|\boldsymbol{s}_{-j})} \mathbb{E}_{\boldsymbol{m}'\sim p(\cdot|\boldsymbol{s})} \left(\mathcal{L}(\boldsymbol{m}) - \mathcal{L}(\boldsymbol{m}')\right)^2\right) \left(\boldsymbol{s}_j^{2(\alpha-1)}(1 - \boldsymbol{s}_j)^{2(\alpha-1)}(\boldsymbol{m}_j - \boldsymbol{s}_j)^2\right)$$
$$\underset{(7)}{\leq} V_{\max}(\boldsymbol{s}) \mathbb{E}_{\boldsymbol{m}_j\sim p(\cdot|\boldsymbol{s}_j)} \left(\boldsymbol{s}_j^{2(\alpha-1)}(1 - \boldsymbol{s}_j)^{2(\alpha-1)}(\boldsymbol{m}_j - \boldsymbol{s}_j)^2\right) \tag{9}$$
$$= \left(\boldsymbol{s}_j^{2\alpha}(1 - \boldsymbol{s}_j)^{(2\alpha-1)} + \boldsymbol{s}_j^{2\alpha-1}(1 - \boldsymbol{s}_j)^{2\alpha}\right) V_{\max}(\boldsymbol{s}). \tag{10}$$

Thus $\mathbb{E}_{\boldsymbol{m}\sim p(\cdot|\boldsymbol{s})} \mathbb{E}_{\boldsymbol{m}'\sim p(\cdot|\boldsymbol{s})} \|\mathcal{G}^\alpha(\boldsymbol{m}, \boldsymbol{m}'|\boldsymbol{s})\|_2^2$ can be estimated as follows:

$$\mathbb{E}_{\boldsymbol{m}\sim p(\cdot|\boldsymbol{s})} \mathbb{E}_{\boldsymbol{m}'\sim p(\cdot|\boldsymbol{s})} \|\mathcal{G}^\alpha(\boldsymbol{m}, \boldsymbol{m}'|\boldsymbol{s})\|_2^2$$
$$= \sum_{j\in\mathcal{C}} \mathbb{E}_{\boldsymbol{m}\sim p(\cdot|\boldsymbol{s})} \mathbb{E}_{\boldsymbol{m}'\sim p(\cdot|\boldsymbol{s})} \left(\mathcal{G}_j^\alpha(\boldsymbol{m}, \boldsymbol{m}'|\boldsymbol{s})\right)^2$$
$$\leq V_{\max}(\boldsymbol{s}) \sum_{j\in\mathcal{C}} \boldsymbol{s}_j^{2\alpha}(1 - \boldsymbol{s}_j)^{(2\alpha-1)} + \boldsymbol{s}_j^{2\alpha-1}(1 - \boldsymbol{s}_j)^{2\alpha}. \tag{11}$$

Thus, when $\alpha \in [\frac{1}{2}, 1)$, we have

$$\mathbb{E}_{\boldsymbol{m}\sim p(\cdot|\boldsymbol{s})} \mathbb{E}_{\boldsymbol{m}'\sim p(\cdot|\boldsymbol{s})} \|\mathcal{G}^\alpha(\boldsymbol{m}, \boldsymbol{m}'|\boldsymbol{s})\|_2^2$$
$$\leq V_{\max}(\boldsymbol{s}) \sum_{j\in\mathcal{C}} \boldsymbol{s}_j^{2\alpha}(1 - \boldsymbol{s}_j)^{(2\alpha-1)} + \boldsymbol{s}_j^{2\alpha-1}(1 - \boldsymbol{s}_j)^{2\alpha}$$
$$\leq |\mathcal{C}| V_{\max}(\boldsymbol{s}).$$

The last inequality holds since the term $\boldsymbol{s}_j^{2\alpha}(1 - \boldsymbol{s}_j)^{(2\alpha-1)} + \boldsymbol{s}_j^{2\alpha-1}(1 - \boldsymbol{s}_j)^{2\alpha}$ is monotonically decreasing w.r.t. $\alpha \in [\frac{1}{2}, 1)$.

Therefore, from Property 1 and 2, we can see that the variance is bounded for any $\boldsymbol{s}$.

$\square$

**Remark 3.** *Eqn. (8) and (9) indicate that $H^\alpha(\boldsymbol{s})$ is introduced to reduce the variance of the stochastic PGE term $\nabla_{\boldsymbol{s}} \ln p(\boldsymbol{m}|\boldsymbol{s})$. Without $H^\alpha(\boldsymbol{s})$ (i.e., $\alpha = 0$), from Eqn.(11), we can see that the total variance bound would be*

$$V_{\max}(\boldsymbol{s}) \sum_{j\in\mathcal{C}} \frac{1}{(1 - \boldsymbol{s}_j)} + \frac{1}{\boldsymbol{s}_j}.$$

*Because of the sparsity constraints, lots of $\boldsymbol{s}_j$ would be close to $0$. Hence, the total variance in this case could be very large.*

**Remark 4.** *Our preconditioning matrix $H^\alpha(\boldsymbol{s})$ plays a role as adaptive step size. The hyperparameter $\alpha$ can be used to tune its effect on variance reduction. For a large variance $\nabla_{\boldsymbol{s}} \ln p(\boldsymbol{m}|\boldsymbol{s})$ we can use a large $\alpha$. In our experiments, we find that simply letting $\alpha = \frac{1}{2}$ works well.*

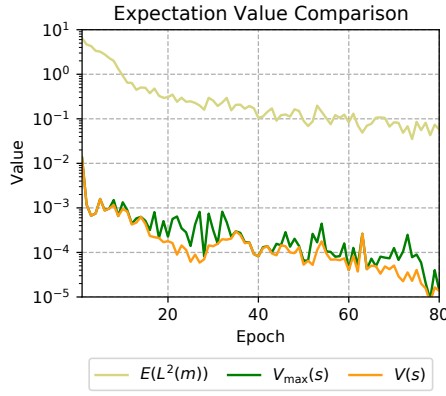

Figure 4: Experiments on ResNet32 on CIFAR-10. $V(\boldsymbol{s})$ and $V_{\max}(\boldsymbol{s})$ are very close during the whole training process and they are smaller than $\mathbb{E}_{\boldsymbol{m}\sim p(\cdot|\boldsymbol{s})}\mathcal{L}^2(\boldsymbol{m})$ by four orders of magnitude.

### A.3  Convergence of Our Method

For the weight update, the convergence can be guaranteed since we use the standard stochastic gradient descent with the gradient calculated via backward propagation.

For the parameter $\boldsymbol{s}$, as stated in Section 4.2.2, we update it as:

$$\boldsymbol{s} \leftarrow \boldsymbol{s} - \eta\left(\mathcal{L}\left(\boldsymbol{m}\right) - \mathcal{L}\left(\boldsymbol{m}'\right)\right) H^\alpha(\boldsymbol{s})\nabla_{\boldsymbol{s}}\ln p(\boldsymbol{m}|\boldsymbol{s}). \tag{12}$$

Let $\Delta\boldsymbol{s}(\boldsymbol{m},\boldsymbol{m}'|\boldsymbol{s})$ be $\left(\mathcal{L}\left(\boldsymbol{m}\right) - \mathcal{L}\left(\boldsymbol{m}'\right)\right) H^\alpha(\boldsymbol{s})\nabla_{\boldsymbol{s}}\ln p(\boldsymbol{m}|\boldsymbol{s})$, we can have

$$
\begin{aligned}
&\mathbb{E}_{\boldsymbol{m}\sim p(\cdot|\boldsymbol{s})}\mathbb{E}_{\boldsymbol{m}'\sim p(\cdot|\boldsymbol{s})}\Delta\boldsymbol{s}(\boldsymbol{m},\boldsymbol{m}'|\boldsymbol{s})\\
=&\mathbb{E}_{\boldsymbol{m}\sim p(\cdot|\boldsymbol{s})}\mathbb{E}_{\boldsymbol{m}'\sim p(\cdot|\boldsymbol{s})}\left(\mathcal{L}\left(\boldsymbol{m}\right) - \mathcal{L}\left(\boldsymbol{m}'\right)\right) H^\alpha(\boldsymbol{s})\nabla_{\boldsymbol{s}}\ln p(\boldsymbol{m}|\boldsymbol{s})\\
=&\mathbb{E}_{\boldsymbol{m}\sim p(\cdot|\boldsymbol{s})}\mathcal{L}\left(\boldsymbol{m}\right) H^\alpha(\boldsymbol{s})\nabla_{\boldsymbol{s}}\ln p(\boldsymbol{m}|\boldsymbol{s}) - \mathbb{E}_{\boldsymbol{m}\sim p(\cdot|\boldsymbol{s})}\mathbb{E}_{\boldsymbol{m}'\sim p(\cdot|\boldsymbol{s})}\mathcal{L}\left(\boldsymbol{m}'\right) H^\alpha(\boldsymbol{s})\nabla_{\boldsymbol{s}}\ln p(\boldsymbol{m}|\boldsymbol{s})\\
=&H^\alpha(\boldsymbol{s})\mathbb{E}_{\boldsymbol{m}\sim p(\cdot|\boldsymbol{s})}\mathcal{L}\left(\boldsymbol{m}\right)\nabla_{\boldsymbol{s}}\ln p(\boldsymbol{m}|\boldsymbol{s}) - H^\alpha(\boldsymbol{s})\mathbb{E}_{\boldsymbol{m}'\sim p(\cdot|\boldsymbol{s})}\mathcal{L}\left(\boldsymbol{m}'\right)\underbrace{\mathbb{E}_{\boldsymbol{m}\sim p(\cdot|\boldsymbol{s})}\nabla_{\boldsymbol{s}}\ln p(\boldsymbol{m}|\boldsymbol{s})}_{I}\\
=&H^\alpha(\boldsymbol{s})\mathbb{E}_{\boldsymbol{m}\sim p(\cdot|\boldsymbol{s})}\mathcal{L}\left(\boldsymbol{m}\right)\nabla_{\boldsymbol{s}}\ln p(\boldsymbol{m}|\boldsymbol{s})\\
=&H^\alpha(\boldsymbol{s})\nabla_{\boldsymbol{s}}\mathbb{E}_{\boldsymbol{m}\sim p(\cdot|\boldsymbol{s})}\mathcal{L}\left(\boldsymbol{m}\right),
\end{aligned}
\tag{13}
$$

where Eqn.(13) holds since term $I = \nabla_{\boldsymbol{s}}\mathbb{E}_{\boldsymbol{m}\sim p(\cdot|\boldsymbol{s})}\mathbf{1} \equiv 0$.

Therefore, we can see that $\Delta\boldsymbol{s}(\boldsymbol{m},\boldsymbol{m}'|\boldsymbol{s})$ is an unbiased gradient estimator associated with an adaptive step size, i.e., our VR-PGE is a standard preconditioned stochastic gradient descent method. Thus, the convergence can be guaranteed.

### A.4  Experiments Verfiying Properties 1 and 2 in A.1

Figure 4 presents the values of $\mathbb{E}_{\boldsymbol{m}\sim p(\cdot|\boldsymbol{s})}\mathcal{L}^2(\boldsymbol{m})$, $V(\boldsymbol{s})$ and $V_{\max}(\boldsymbol{s})$ during the training process of ResNet-32 on CIFAR-10. We can see that $V(\boldsymbol{s})$ and $V_{\max}(\boldsymbol{s})$ are very close during the whole training process and they are smaller than $\mathbb{E}_{\boldsymbol{m}\sim p(\cdot|\boldsymbol{s})}\mathcal{L}^2(\boldsymbol{m})$ by four orders of magnitude. This verifies our Property 1 and 2.

## B  Experimental Configurations

**[CIFAR-10/100 Experiments]** GPUs: 1 for VGG and ResNet and 2 for WideResNet. Batch Size: 256. Weight Optimizer: SGD. Weight Learning Rate: 0.1. Weight Momentum: 0.9. Probability Optimizer: Adam. Probability Learning Rate: **12e-3**. WarmUp: ✗. Label Smoothing: ✗.

**[ImageNet-1K Experiments]** GPUs: 4. Batch Size: 256. Weight Optimizer: SGD. Weight Learning Rate: 0.256. Weight Momentum: 0.875. Probability Optimizer: Adam. Probability Learning Rate: **12e-3**. WarmUp: ✓. Label Smoothing: 0.1.

**Remark 5.** *The bold-face probability learning rate 12e-3 is the **only** hyperparameter obtained by grid search on CIFAR-10 experiments and applied directly to larger datasets and networks. Other hyperparameters are applied following the same practice of previous works [34, 20, 27, 51]. The channels of ResNet32 for CIFAR experiments are doubled following the same practice of [42].*

Table 5: Forward/backward time of dense/sparse networks and accompanying properties.

| Model | Val Acc(%) | Params(%) | Forward(min) | Backward(min) | Train-Computational Time(min) |
|---|---|---|---|---|---|
| | 93.84 | 100.00 | 6.89 | 14.96 | 21.85 (1.00×) |
| | 93.46 | 23.71 | 6.41 | 7.63 | 14.04 (1.55×) |
| | 93.11 | 12.75 | 4.89 | 5.54 | 10.43 (2.09×) |
| VGG-19 | 92.23 | 6.69 | 3.10 | 3.73 | 6.83 (3.20×) |
| | 90.82 | 3.06 | 2.15 | 2.71 | 4.86 (4.50×) |
| | 87.97 | 0.80 | 1.27 | 1.68 | 2.95 (7.41×) |

## C  Calculation Schemes on Train-cost Savings and Train-computational Time

### C.1  Train-cost Savings

The train-cost of vanilla dense training can be computed as two parts: in forward propagation, calculating the loss of weights and in backward propagation, calculating the gradient of weights and gradient of the activations of the previous layers. The FLOPs of backward propagation is about 2∼3 times of forward propagation [2]. In the following calculation, we calculate the FLOPs of forward propagation concretely and consider FLOPs of backward propagation 2 times of forward propagation for simplicity.

**[GrowEfficient]** The forward propagation of dense network is $f_D$. The forward propagation of GrowEfficient is partially sparse with FLOPs being $f_S$ and backward propagation is dense. Therefore the train-cost saving is computed as $\frac{f_D+2f_D}{f_S+f_D} = \frac{3}{2+f_S/f_D}$, upper-bounded by $\frac{3}{2}$.

**[Ours]** The forward propagation of dense network is $f_D$. The forward propagation and backward propagation is totally sparse. The FLOPs of forward propagation is $f_S$ and the FLOPs of backward propagation is $2*f_S$. The forward propagation has to be computed two times. Therefore the train-cost saving is computed as $\frac{f_D+2*f_D}{2*f_S+2*f_S} = \frac{3}{4f_S/f_D}$. Actually, $f_S/f_D$ is roughly equal to $\rho^2$, leading to drastically higher train-cost savings.

### C.2  Train-computational Time

The calculation of train-computational time focuses on the forward and backward propagation of dense/sparse networks. For both of the dense and sparse networks, we sum up the computation time of all the forward and backward propagation in the training process as the train-computational time. The detailed time cost is presented in Table 5. We can see that we can achieve significant speedups in computational time.

## D  Potentials and Limitations of This Work

**[On Computational Cost Saving]** Although our method needs two forward propagation in each iteration, we have to point out that our method can achieve significant computational cost saving. The reason is that our forward and backward is completely sparse, whose computational complexity is roughly $\rho^2 * 100\%$ of the conventional training algorithms with $\rho$ being the remain ratio of the channels.

**[On Exploring Larger Networks]** About the potential of our method in exploring larger networks, we'd like to clarify the following three things:

1. The memory cost of the structure parameters $s$ is negligible compared with the original weight $w$ as each filter is associated with only one structure parameter, therefore our $s$ would hardly increase the total memory usage.

2. Although in our method, we need to store the parameter of the full model, this would not hinder us from exploring larger networks. The reason is that, in each iteration, we essentially perform forward and backward propagation on the sparse subnetwork. More importantly, we find that reducing the frequency of sampling subnetwork, e.g., sample a new subnetwork for every 50 iterations, during training would not affect the final accuracy. In this way, we can store the parameters of the full model on CPU memory and store the current subnetwork on GPU, and synchronize the parameters' updates to the full model only when we need to resample a new subnetwork. Hence, our method has great potentials in exploring larger deep neural networks. We left such engineering implements as the future work and we also welcome the engineers in the community to implement our method more efficiently.

3. In exploring larger networks, the channel remain ratio $\rho$ can be much smaller than the one in the experiments in the main text. Notice that our method can reduce the computational complexity to $\rho^2 * 100\%$ of the full network. It implies that, in this scenario, the potential of our method can be further stimulated. We left this evaluation as future work after more efficient implementation as discussed above.