# OpenReview forum: "Efficient Neural Network Training via Forward and Backward Propagation Sparsification"
_NeurIPS.cc/2021/Conference — NeurIPS 2021 Poster_

### Official Review · Reviewer_CRsd · 2021-07-13

**Rating:** 4
**Confidence:** 4

**Summary:**

This paper addresses the topic of sparse training of neural networks. They target channel-level sparsity with the goal of realizing practical training speedups with existing software and hardware. The authors study the limitations of existing techniques in terms of their ability to exploit sparsity during training and show a wide array of empirical results with their proposed technique.

**Ethical Concerns:**

There are not ethical issues with this paper.

**Limitations And Societal Impact:**

I did not identify potential negative societal impacts in this work.

**Main Review:**

The framing of this paper and it’s “truly sparse neural network training” technique is sensational and does not acknowledge the contributions of prior work on this topic.

Firstly, It is not correct to say that no existing method can effectively accelerate training. There is a body of literature on sparse training that has developed techniques for exactly this. For example: SET [5], DeepR [6], SNFS [7], RigL [2] and Top-KAST [3]. The authors cite some of these techniques but don’t seem to acknowledge their contributions.

Second, it is not correct to say that sparse matrix multiplication cannot be implemented efficiently on widely-used platforms. Recent work has shown success in doing so for sparse deep learning workloads [8, 9].

Third, when discussing channel-sparse algorithms, the authors state “It is obvious that these methods cannot achieve significant speedup in training since they need to calculate the full gradient in backward propagation”. This statement is an exaggeration. Even if the weight gradient computation needs to produce a dense gradient w.r.t. the sparsified weights the forward pass and backward gradient computation w.r.t. the input data to the linear operation can still take advantage of sparsity. That is to say 2/3rds of the FLOPs in the model still benefit from sparsity. This discrepancy leads me to believe that the authors could be underestimating the training-time FLOPs saving for other techniques in their experiments.

Delving into this, the only existing technique for which the authors provide “training-cost savings” data in their experiments (Tables 2 & 3) is GrowEfficient. In their response, I’d like the authors of this work to clarify how they calculate the “training-cost savings” for GrowEfficient. Even if the weight gradients are calculated densely, the forward pass and half of the backward pass should benefit from sparsity.

On the topic of weight gradient computation, it’s not clear to me that GrowEfficient cannot exploit sparsity in these as well. As I understand it, GrowEfficient uses a similar Bernoulli random variable (q) on their weight matrices to induce sparsity. For cases where this value is 0, any gradient for the zeroed neuron should also be zero and thus GrowEfficient should also be able to exploit sparsity in their weight gradient computation. The authors of this work stated that “The authors of GrowEfficient [46] confirmed that actually they also calculated the gradient of qc w.r.t, sc in their Eqn.(6) via STE even if qc = 0.”. If this is true it would be correct to count the cost of computing a dense gradient for GrowEfficient. However, I was unable to confirm this from the GrowEfficient paper and basing conclusions on hearsay that is not documented in the literature feels tenuous. The clearest way to demonstrate this is true would be for the authors to re-implement the baseline technique, although I realize this is a considerable burden.

Other Issues
- In Table 1, many of these techniques are not “sparse training” algorithms, i.e. their aim is not to accelerate training with sparsity. For example, references 7, 50 and 28. Reference 19 is listed under both parametric and non-parametric techniques.
- In general, the taxonomy of “non-parametric” and “parametric” sparsification methods is unclear and appears to be limited in scope. For example, it does not take into account details of the algorithm captured by existing taxonomies like “dense-to-sparse” v. “sparse-to-sparse”. I’d recommend the authors take a look at the taxonomies used in recent literature relating to sparsity. For example, Gale et. al [1], Evci et. al [2], Jayakumar et. al [3], and Hoefler et. al [4]. As I understand it, “parametric” techniques use some form of sparsity inducing regularization and “non-parameteric” techniques do something else. I don’t think a new term is necessary to make this distinction.
- Ideally the authors would include training-cost savings for more techniques than GrowEfficient in their experiments. Given these theoretical savings rarely translate directly into runtime savings, it would improve the paper greatly if the authors included runtimes for more techniques than their own in Table 4.

References
[1] https://arxiv.org/abs/1902.09574
[2] https://arxiv.org/abs/1911.11134
[3] https://arxiv.org/abs/2106.03517
[4] https://arxiv.org/abs/2102.00554
[5] https://www.nature.com/articles/s41467-018-04316-3
[6] https://arxiv.org/abs/1711.05136
[7] https://arxiv.org/abs/1907.04840
[8] https://arxiv.org/abs/1911.09723
[9] https://arxiv.org/abs/2006.10901

**Time Spent Reviewing:**

3hrs

---

> ### Author Response · Authors · 2021-08-10
> **Reply to Reviewer CRsd (2)**
>
> **Q6:  How to calculate the training-cost savings for GrowEfficient?**
>
> **A6:** It has been clarfied in Section C.1 of the appendix. It is metioned in line 253-254 in maintext.
>
>
> **Q7: GrowEfficient uses a dense backpropagation procedure.**
>
> **A7:** The backpropagation for computing the  **gradients of weights** can be sparse, following the same logic as Section 4.2.1. The dense part of computation comes from the dense backpropagation to compute the **gradients of structure parameters**. That is, from equation (3), we find that the computation of gradients of structure parameters involve dense backpropagation process and partially sparse forward propagation process. It is not an issue of implementation but an issue of algorithm: if the gradients of $q_{c}$ with $s_{c}=0$ are not calculated to update $q_c$, GrowEfficient will collapse.  We present experimental results as follows:
>
> |  | Top-1 Acc(%) | FLOPs (%) |
> | ----------- | ----------- | ----------- |
> | GrowEfficient (update all structure parameters)| **81.8**        | 1.60 |
> | | **75.98**       | 0.99 |
> | GrowEfficient (update structure parameters with mask = $1$)|  10.0      |      3.69         |
> | |       10.0      |      1.38        |
>
>
> The reported result of GrowEfficient cannot be reproduced by only updating structure parameters with mask = $1$.
>
> Intuitively, we can understand it from a simple case. In Eqn.(6) of GrowEfficient (ICLR2021 version), suppose that some channel $c$ has a very very small score $s_c$ and the corresponding mask $q_c=0$. On one hand, if we do not calculate the gradient of $s_c$ and update it in backward propagation, then $s_c$ would stay small during the whole training process. This can make $q_c$ always be 0 and behave like a deterministic/dead mask, easily making the training failed. On the other hand, if the loss can change significantly by flipping $q_c$, we can expect that the gradient of $s_c$ is nonzero. Therefore, in fact, the gradient of $s_c$ can be nonzero even if $q_c=0$. Hence, if we calculate the gradient in backward propagation, the computation has to be dense.
>
>
> You may double-check with the authors with the update rule through email.
>
>
> **Q7: In Table 1, many of these techniques are not “sparse training” algorithms.**
>
> **A7:** They are listed to demonstrate the performance of our method in pruning. Our method can achieve even better result of final sparse model with significant speed up than previous pruning methods. We do not find any other strong sparse training baselines besides GrowEfficient.
>
> **Q8:  Reference 19.**
>
> **A8:** Reference 19 is parametric method. It is a typo.
>
> **Q9: In general, the taxonomy of “non-parametric” and “parametric” sparsification methods is unclear and appears to be limited in scope.**
>
> **A9:** We are the first to use the taxonomy of “parametric” and “non-parametric” to classify the existing methods. These two classes of methods calculate the gradients and learn the sparse structure in different ways. Parametric methods ultilize structure parameters as the auxiliary to distinguish the importance of weights/channels. Non-parametric methods don't add  structure parameters in the training process and ultilize only the properties of weights themselves to distinguish the importance of weights/channels, like weight magnitudes, gradient magnitudes. To differentiate from the definitions of parametric/non-parametric in some machine learning methods, we will modify the terminology to avoid confusion.
>
>
>
> **Q10: Ideally the authors would include training-cost savings for more techniques than GrowEfficient in their experiments.**
>
> **A10:** GrowEfficient is the SOTA method on sparse training. To the best of our knowledge, no significant speedups in training are reported in the literature. We will include some baselines such as SNFS which can achieve some practical speedup by careful implementation, although they are not stronger than GrowEfficient. If you find some exising methods demonstrated with significant practical speedups (e.g., larger than 3) in training, please tell us and we are willing to compare with them.
>
> **Q11: It would improve the paper greatly if the authors included runtimes for more techniques than their own in Table 4.**
>
> **A11:** To the best of our knowledge, no significant speedups in training time are reported in the literature. If you find some existing methods demonstrated with both high accuracy and significant practical  training time speedups (e.g., larger than 3) in training, please tell us and we are willing to compare with them.

---

> > ### Comment · Reviewer_CRsd · 2021-08-13
> > **Reviewer Response**
> >
> > I disagree that “truly sparse neural network training” should be defined as you have said. For conditions 1 & 2, it’s fine for a sparse training algorithm to do some dense computation. Algorithms like RigL or Top-KAST do this and amortize that dense computation over many steps of sparse computation. The memory usage of doing dense gradient computation (e.g., in RigL) isn’t bad because the matrices can be materialized one at a time (i.e., calculate dense gradient, perform update, free memory, repeat for next layer). I don’t think condition 3 is reasonable - it is possible to implement sparse training in PyTorch/TensorFlow. The fact that the proper kernels aren’t integrated in the frameworks should not affect the definition of “sparse neural network training”. The only condition of claiming to be doing “sparse neural network training” is that you have sparsity present in your training process.
> >
> > I understand the author’s are trying to make a distinction about the practical utility of their approach. This is a good thing to highlight about their approach, but I think the terminology they’ve chosen is poor in that they’ve re-defined “sparse neural network training” rather than explaining the practicality of their approach in some other way.
> >
> > Q3: The kernels released with [9] support sparse gradient computation in TensorFlow. You can see the code on GitHub [2].
> >
> > Q4: The upper bound of 3 makes sense, but I’d much rather see the authors show this quantitatively rather than discounting methods because they do not meet the author’s definition of “significant”. The author’s point about not being able to implement this in TensorFlow or PyTorch is incorrect. There are a number of third party libraries for sparse computation in TensorFlow and PyTorch (For example, Triton [1], SGK [2], pytorch_sparse [3]).
> >
> > I also don’t think defining “significant” to be some specific threshold is clear. I’d prefer to see the author’s explain what they mean when discussing the potential for speedups of different methods.
> >
> > Some of the confusion around the framing of this paper stems from the fact that this paper is focused on channel/neuron sparsity (as opposed to unstructured sparsity). I think the paper might be more clear if the author’s re-framed the text to make this very clear, rather than using the phrasing of “sparse training” that evokes the fully general form of sparsity.
> >
> > I’ve raised my score based on the author’s responses w.r.t. GrowEfficient benchmarking. If the author’s main claim is practical training speedups from channel sparsity I’d still like to see more thorough empirical results. For example, actual training run times rather than just theoretical training cost reductions for more models/datasets. I’d also like to see training run times for GrowEfficient. Ideally, the authors would include more baselines (e.g., those suggested by reviewer T3oD).
> >
> > References
> > [1] https://github.com/openai/triton
> > [2] https://github.com/google-research/google-research/tree/master/sgk
> > [3] https://github.com/rusty1s/pytorch_sparse

---

> > > ### Author Response · Authors · 2021-08-16
> > > **Reply to Reviewer CRsd (4)**
> > >
> > > **Q19:**  "The upper bound of 3 makes sense, but I’d much rather see the authors show this quantitatively rather than discounting methods because they do not meet the author’s definition of significant."
> > >
> > > **A19:**
> > >
> > > As shown in A16, PyTorch/Tensorflow don't support the training of general sparse convolutional neural networks. While we could achieve some speed-up with frozening the mask by the criterion in SoftNet, i.e. (L2 norm filter selection) to virtually do hard pruning in late epochs (after norms are more stable)(Actually this is the way how the authors of GrowEfficient calculate the speed-up of SoftNet). We compare with SoftNet[13] on ResNet-20 on Cifar-10. Here is the result:
> > >
> > > | Model | Method | Top-1 Acc(%) |Params(%)| FLOPs(%) | Train-cost Savings($\times$)
> > > | ----------- | ----------- | ----------- |  ----------- | ----------- | ----------- |
> > > | VGG-19 | Original | 92.9 |100.00 |100.00       | 1.00 |
> > > |  | SoftNet | 92.1 | 36      | 36.1 | 1.6 |
> > > |  | Ours | **92.5** |**4.4**| **8.7** | **8.69** |
> > >
> > > It shows that the train-cost saving of SoftNet is largely below 3 even in this careful implementation, validating our claim that SoftNet cannot speed-up the training process significantly.
> > >
> > > **Q20:** Experimental results on practical training speedups on more datasets/models and comparison against GrowEfficient.
> > >
> > > **A20:**
> > >
> > > First we present the comparison of our method and GrowEfficient on VGG-19 on CIFAR-10.
> > >
> > > | Model | Method | Top-1 Acc(%) | FLOPs(%) | Train-cost Savings($\times$) | Train-computational Time Savings ($\times$)|
> > > | ----------- | ----------- | ----------- |  ----------- | ----------- | ----------- |
> > > | VGG-19 | Original | 93.84 |100.00       | 1.00 | 1.00|
> > > |  | GrowEfficient | 92.3 | 27.28       | 1.19 | 1.11 |
> > > |  | GrowEfficient | 91.68 |19.90       | 1.22 | 1.13 |
> > > |  | GrowEfficient | 91.08 | 14.02      | 1.26 | 1.15 |
> > > |  | GrowEfficient | 90.54 | 9.12      | 1.31 | 1.18 |
> > > |  | GrowEfficient | 87.61 | 3.08      | 1.37 | 1.25 |
> > > |  | Ours | **93.46** | **28.57**       | **2.64** | **1.55** |
> > > |  | Ours | **93.11** | **19.33**       | **3.89** | **2.09** |
> > > |  | Ours | **92.23** | **10.27**      | **7.30** | **3.20** |
> > > |  | Ours | **90.82** | **4.94**      | **15.28** | **4.50** |
> > > |  | Ours | **87.97** | **1.70**      | **44.68** | **7.41** |
> > >
> > > Next we present the comparison of our method and GrowEfficient on ResNet-50 on ImageNet.
> > >
> > > | Model | Method | Top-1 Acc(%) | Params(%)  | FLOPs(%) | Train-cost Savings($\times$) | Train-computational Time Savings ($\times$)|
> > > | ----------- | ----------- | ----------- | ----------- | ----------- | ----------- | ----------- |
> > > | ResNet-50 | GrowEfficient | 75.2 |61.2 | 50.3       | 1.10 | 1.075 |
> > > |  | GrowEfficient | - |11.3 | 10.8       | 1.28 | 1.16 |
> > > |  | Ours | **76.0** |**48.2** | **46.8**       | **1.60** | **1.33** |
> > > |  | Ours | **69.3** |**10.8** | **10.1**       | **7.36** | **3.2** |
> > >
> > > "-" means that GrowEfficient collapses at such high pruning rate, 11.3% remaining parameters. The reason may come from the inaccurate gradient estimation by straight through approximator. We find that our method beats GrowEfficient steadily both in terms of train-cost savings and train-computational time savings. The gap between our method and GrowEfficient becomes larger as the pruning rate goes higher.
> > >
> > > **Q21:** Comparison against (1) ClickTrain, PruneTrain and (2) DMCP, MetaPruning (two pruning after training methods).
> > >
> > > **A21:**
> > > **(1)** For the comparison against ClickTrain and PruneTrain, it is observed that PruneTrain and ClickTrain work with a large model at the begining epochs and consume huge memory and heavy computation in the early stage. The results (Table 1 of PruneTrian and Fig. 12 of ClickTrain) show that the speedup ClickTrain can achieve is close to 1 and PruneTrain cannot preserve the accuracy well although it can achieve some speedup. We will add these two baselines to better demonstrate the superiority of our method while GrowEfficient is already the SOTA method for comparison.
> > >
> > > **(2)** For the comparison against DMCP and MetaPruning, DMCP and MetaPruning are pruning methods with no speed-up. The comparison of final pruned network is presented as follows:
> > >
> > > | Method  | Top-1 Acc(%) | FLOPs (%) | Train-cost Savings($\times$)|
> > > | ----------- | ----------- | ----------- | ----------- |
> > > | MetaPruning| 75.4        | 48.9 | -|
> > > | DMCP    |       **76.2**        |      53.7         | - |
> > > | Ours   | 76.0        | **46.8** | **1.6**|
> > >
> > > We find that our method obtains 1.6$\times$ training speed-up while achieving accuracy and FLOPs better than MetaPruning and comparable results against DMCP.
> > >
> > > We hope our responses above assist in resolving your concerns and better re-evaluating our paper.

---

> > > ### Author Response · Authors · 2021-08-16
> > > **Reply to Reviewer CRsd (3)**
> > >
> > > Thanks for your prompt reply and re-evaluating our paper. Below we present the answers to your concerns.
> > >
> > > **Q12: The rigor of the definition of "truly sparse".**
> > >
> > > **A12:** We agree with you that our definition is not so rigorous. We set these criteria since they can guarantee significant practical speedup and we do not intend to develop a new rigorous definition for the community. Moreover, developing a definition for sparse training as rigorous as some mathematics definitions (e.g., linear space, matrix product) is very difficult and even impossible. Actually, the contribution of this paper is our efficient training algorithm instead of the definition. Thus we think it is not a fundamental issue. And to solve it, we will change our statement in the final version to indicate that these are the distinctions/advantages of our method and we do not intend to propose a new rigorous definition in the community.
> > >
> > > **Q13: Some dense computations are allowed.**
> > >
> > > **A13:** Yes, we agree with you. But the important thing is that such dense computation should not take up a large amount in the overall training process and also should not significantly increase the memory usage.
> > >
> > > Moreover, we would like to point out that, for convolutional networks,  the dense computation for pruning in RigL would increase the memory usage to almost that of the dense model. The reason is that for convolutional networks, storing the activations for a mini-batch takes up most of the memory usage. We need to store all the activations before we start the backward propagation. Therefore, layer by layer gradient computation cannot alleviate this limitation.
> > >
> > > At last, we would like to say that the limitation of RigL and Top-Kast is not their dense computation for update the mask. Their limitation is that they are weight-level methods, which are not supported by Tensorflow and Pytorch because implementing the backward propagation of general sparse convolution is difficult. The detailed reason is given in A16.
> > >
> > > **Q14: The memory usage of doing dense gradient computation (e.g., in RigL) isn’t bad because the matrices can be materialized one at a time (i.e., calculate dense gradient, perform update, free memory, repeat for next layer).**
> > >
> > > **A14:** For convolutional networks, such kind of computation cannot alleviate this limitation. The reason is that for convolutional networks, storing the activations for a mini-batch takes up most of the memory usage. We need to store all the activations before we start backward propagation. To be honest, such limitation can be alleviated by another kind of clumsy implement, that is, in backward propagation for each layer, run forward propagation for one more time to obtain the activations needed to the backward propagation in that layer and free all other activations. In this way, for a L-layer network, we need to run forward propagation for L times to complete a single backward propagation, which could be costly.
> > >
> > >
> > >
> > > **Q15: About condition 3.**
> > >
> > > **A15:** Actually, the libraries such as  Triton [1], SGK [2] and  pytorch-sparse [3] can only be used to implement the backward propagation for sparse matrix multiplication instead of general convolutional operator. Please refer to A16 for more details. We also cannot find empirical evidences showing that they can accelertate the backward propagation of general sparse convolution.
> > >
> > >
> > > **Q16: The kernels released with [9] support sparse gradient computation in TensorFlow. You can see the code on GitHub [2].**
> > >
> > > **A16:** Actually [9] can only support the backward propagation of sparse matrix multiplication but not general sparse convolution. The backward propagation of sparse convolution is more complex than that of general sparse matrix multiplication. In page 10 of [9], the authors give the empirical results on MobileNetV1, but it does not indicate that [9] supports general sparse convolution. The evidences are:
> > >
> > > **1)** As the authors stated in the experimental setup in the right column of page 10 of [9], 'We introduce sparsity into the **1×1
> > > convolutions** of MobileNetV1 (not 3×3 depthwise convolution) using magnitude pruning'. The reason is that 1×1 convolution is essentially a matrix-vector product, just as the authors stated in the end of right column of page 10, i.e, "The 1×1 convolutions in these models are ... and **can be computed as matrix multiplication** if the input data is stored in CHW format". To understand it, we can vectorize each channel in a layer and group them into a matrix, then the 1x1 convolution is exactly a standard matrix-vector product.
> > >
> > > **2)** The authors also stated in "Results \& Analysis" in the right column of page 10, "the **depthwise convolutions** become a **significant bottleneck** after the 1x1 convolutions are pruned".
> > >
> > > That's why [10] only gives the results on MobileNetV1 but not other convolutional networks such as ResNet/VGG. Morevover, as the authors stated in "experimental setup" in the right column of page 10, for MobileNetV1, they mainly "target efficient inference in the regime where inference costs outweigh training costs, we **increase training time** for our sparse models **by 10×** which helps the sparse models converge while being pruned". Therefore, the practical speedup for training is still unclear. Although the problem of sparse backward propagation for convolution maybe solved in the future, to achieve practical speedup on current platforms pytorch and tensorflow is meaningful at present, which is the main contribution of this work.
> > >
> > > **On the other hand, we can obtain the answer by simple reasoning: if the weight-level sparse training is already well-supported by PyTorch/Tensorflow, there is no need to develop channel-level sparse training method since weight-level can be more efficient on reducing the model size and computational cost. However, this is obviously not the fact.**
> > >
> > > **Q17: I don’t think defining “significant” to be some specific threshold is clear.**
> > >
> > > **A17:** Yes, strictly speaking, it is not reasonable to set a specific threshold for a qualitative variable. But we would like to give our understandings. We think that a method with significant speedup should not be constrained with a low upper bound such as 2 or 3, since we can usually achieve this kind of speedup by using more computational resource and thus this usually cannot enable new applications for deep neural networks. In contrast, the train-cost savings of our method is $\frac{3}{4f_S/f_D}$ with $f_S/f_D \approx \rho^2$ and $\rho$ being the remaining channel ratio (see Section C.1 in the appendix for more details), which is much higher and more promising.  As researchers on machine learning and computer science, we think it is good (our mission in some sense) to develop efficient techniques with large potentials (we mean higher upper bounds), which are promising and could make new real applications possible in the future.
> > >
> > > **Q18: "Some of the confusion ... evokes the fully general form of sparsity".**
> > >
> > > **A18:** Yes. Thanks for the reminder. We will clarify this in the final version if accepted.

---

> > > ### Author Response · Authors · 2021-08-28
> > > **Reply to Reviewer CRsd (5)**
> > >
> > > Thanks for your constructive advice. As we stated earlier in the rebuttal, we will follow your advise and we are happy to modify our statements in our paper. Here is what we would like to solve the issue based on your feedbacks:
> > >
> > > **(1)** Replace our title by "Efficient Channel-level Sparse Neural Network Training".
> > >
> > > **(2)** Remove all the words, such as  "first", "truly sparse" and so on in the main text.
> > >
> > > **(3)** Remove all the controversial statements such as "existing methods cannot achieve significant speedups".
> > >
> > > **(4)** Claim in the beginning of this paper that we focus on channel-level pruning to avoid some confusions from weight-level methods.
> > >
> > > **(5)** Instead of giving new definition of 'truly sparse', we will only consider them as good properties of our method, which we will state as follows:
> > >
> > > **(5.1)** The computation in both forward and backward propagations of our method is  completely sparse, i.e., the computational complexity is significantly lower than that in standard training.
> > >
> > > **(5.2)** During the whole training procedure, our method  works on small sub-networks with the target sparsity, i.e., both the forward and backward propagations only need to  go through a small sub-network instead of the dense network.
> > >
> > > **(5.3)** Our method can be implemented easily on the widely-used platforms on Pytorch or Tensorflow.
> > >
> > > **Thanks again for your constructive comments on our work. If you feel this rebuttal resolves all your concerns, we appreciate it if you could raise your ratings. If you have any other concerns, please let us know and we are happy to discuss with you.**

---

> ### Author Response · Authors · 2021-08-10
> **Reply to Reviewer CRsd (1)**
>
> **Q1: The framing of this paper and it’s “truly sparse neural network training” technique is sensational and does not acknowledge the contributions of prior work on this topic.**
>
> **A1:** We think it is not a fundamental issue and it comes from our different understanding about "truly sparse", "significantly accelerate" and "in practice". We certainly acknowledge the contributions of the existing methods while we would like to clarify our understanding/definition about "truly sparse" here and we will also modify our statements in the final version accordingly if accepted. To ensure "significant speedup in practice", we think "truly sparse algorithm" should satisfy the following **three criteria**:
>
> **1) The computation in both forward and backward propagations should be completely sparse, i.e., the computational complexity should be significantly lower than that in standard training (Remark 1).** Especially the gradients to weights should be sparse rather than dense. This excludes algorithms like SWAT, SoftFilter.
>
> **2) During the whole training procedure, the training algorithm should always work on small sub-networks with the target sparsity, i.e., both the forward and backward propagations only need to  go through a small sub-network instead of the dense network (line 233).** The reason why we set this criterion is that both memory usage and computational cost of  such totally sparse training algorithm are bounded  from beginning to the end of training, leading to significant savings in both memory usage and computational cost and leaving broad space for exploring larger models.  This excludes dense-to-sparse training methods like PruneTrain (Lym, Sangkug, et al. 2019) and ClickTrain (Zhang, Chengming, et al., 2021), which consume huge memory and heavy computation in the early stage.
>
> **3) The training algorithm should be able to be implemented easily on the widely-used platforms on Pytorch or Tensorflow, since  most researchers today use these two platforms to train neural networks and explore new models (line 60).** Although many weight-level sparse training methods lead to speed-up in training theoretically, the real implementation of such algorithm involve huge efforts inaccessible for individual researchers or engineers. Therefore, developing easy-to-use sparse training algorithms on these two platforms is meaningful for broad audience. In contrast, our algorithm can be easily implemented on Pytorch and Tensorflow, and we will release our code on acception.
>
> We summerize the previous methods into different groups:
>
> | Weight-level  |  |Channel-level| |
> | ----------- | ----------- | ----------- | ----------- |
> | |  Non-parametric        |  Parametric |  Parametric|
> |   |               |      Previous methods       |Our method |
> | Not applicable to widely-used platforms like PyTorch and Tensorflow, lack of support from sparse backward propagation for convolutional neural networks. | Cannot achieve significant speed-up constrained by calculating dense gradients of weights and dense-to-sparse training paradigm.       |Calculating gradients of structure parameters by dense backward propagation. | Calculating gradients of structure parameters by sparse forward propagation.|
> |Violates criterion 3.    |   Violates criteria 1,2.    |Violates criteria 1.      | The first to satisfy all three criteria. |
>
> To the best of our knowledge, our method is the first to satisfy the three criteria at the same time. We  notice that some existing methods can achieve some speedup in training by careful implementation. For example, a few channels have chances to be removed by weight-level algorithms when the weights in these channels are all pruned. For some dense update algorithms, one can also remove some channels if the corresponding weights are quite small for a long time. However, even with such careful implementations/ hand-crafted tricks, no significant speedups are reported in the literature. As shown in SNFS (Dettmers and Zettlemoyer [2020]), even on the simple classification task on CIFAR10, the speedup one can achieve can only be 1.07-1.32.  Therefore, compared with these methods, our method is more theoretically grounded and can achieve significant speedups in training without such heuristic tricks in implementation.
>
> We hope the clarification above assist in resolving your concerns  and better evaluating our paper.
>
> **Q2:  Exist methods such as SET [5], DeepR [6], SNFS [7], RigL [2] and Top-KAST [3] effectively accelerate training.**
>
> **A2:** We acknowledge their significant contributions. They demonstrated the possibility to compress DNNs and make it possible to deploy DNNs on small devices.  Besides, we would like to point out that these methods are weight-level algorithms and they mainly target on reducing theoretical training FLOPs. No significant practical speedups in training are reported in these papers.
>
> Pytorch and Tensorflow currently do not support weight-level sparse backward propagation. We notice that a few channels have chances to be removed by weight-level algorithms when the weights in these channels are all pruned, which can lead to practical speedup. However, as shown in Table 3 of SNFS[7], even on the simple classification task with CIFAR10, the speedup achieved by such careful implementation can only be 1.07-1.32. The reason is that the number of pruned channels is very limited.
>
> **Q3: It is not correct to say that sparse matrix multiplication cannot be implemented efficiently on widely-used platforms. Recent work has shown success in doing so for sparse deep learning workloads [8, 9].**
>
> **A3:** It is a misunderstanding. In line 56, we mean that Pytorch and Tensorflow do not support weight-level sparse backward propagation for convolutional networks. We will correct this statement.
>
> [8] and [9] are two papers for accelerating inference instead of training. To the best of our knowledge, no empirical evidences are given in the literature that Pytorch and Tensorflow can be used to efficiently implement sparse backward propagation to train convolutional networks.
>
> **Q4: The statement “It is obvious that these methods [13,42] cannot achieve significant speedup in training since they need to calculate the full gradient in backward propagation” is an exaggeration. The forward pass and backward gradient computation w.r.t. the input data to the linear operation can still take advantage of sparsity. That is to say 2/3rds of the FLOPs in the model still benefit from sparsity. "**
>
> **A4:** [13] and [42] mainly aim at pruning the networks to improve inference efficiency.
>
> Firstly, [13] and [42] may ultilize the sparsity of weights during the forward and 1/2 backward training process, and the train-cost saving of pruning process is below 3. But 3 can only be achieved when the model is pruned significantly, that is only occurs at the end part of training (most of channels are pruned) for the dense to sparse methods. So the over all real train-cost saving of whole pruning stage is largely below 3 since they are dense-to-sparse training methods. This is not a "significant" speed-up.
>
> Secondly, the implementation of such update rule is not applicable to widely-used platforms like PyTorch and Tensorflow, making $3$ a theoretical bound instead of a practical bound. The reason is that we cannot sparsify the gradient computing w.r.t the input data/activation on Pytorch and Tensorflow. You can understand it from the following contradiction, i.e., to sparsify the gradient computing w.r.t. the activation we need to remove the zero-valued channels from the graph, while to calculate the gradient w.r.t. the weights of these channels we need to preserve the channels.
>
> **Q5: "This discrepancy leads me to believe that the authors could be underestimating the training-time FLOPs saving for other techniques in their experiments." "On the topic of weight gradient computation, it’s not clear to me that GrowEfficient cannot exploit sparsity in these as well."**
>
> **A5:** There is some misunderstanding between non-parametric and parametric methods. For non-parametric methods, 1/2 of the backpropagation process is sparse. For parametric methods, the backpropagation process can be sparse if we only use it to calculate the gradient of weights.
>
> GrowEfficient can indeed exploit sparsity in weight gradient computation . However, GrowEfficient also uses backpropagation to compute the gradient of the structure parameters, which has to be dense.  We present the experimental results and intuitive understanding in Q7. The calculation method of the training-cost saving of GrowEfficient is described in Section C.1 (line 253-254).
>
> For the methods other than GrowEfficient, they are pruning methods not meant for training speed-up. They require pre-trained models (line 256-261 in our paper) and the train-cost saving is below 1 (SoftNet can only achieve state-of-the-art results with pretrained models as stated in the original paper, page4, left column). You can double-check this calculation method with the original author of GrowEfficient. They confirmed that it is correct to deem these pruning methods as no speed-up exists. Therefore, there is no underestimation of other techniques in our experiment.

---

### Official Review · Reviewer_CTnr · 2021-07-14

**Rating:** 7
**Confidence:** 3

**Summary:**

This paper proposes an efficient sparse training method to solves the problem of accelerating the training speed of deep neural networks and save memory usage.  The authors first formulate the training process as a continuous minimization problem under global sparsity constraints.  Then, separating the optimization process into two steps, corresponding to weight update and structure parameter update. A variance-reduced policy gradient estimator is proposed to update the structure parameter. Finally, the experimental results are solid and strong. The problem has lots of meanings for engineering implementation.

**Limitations And Societal Impact:**

The only limitation of this paper is the complexity of the method implementation, especially for the VR-PGE part. The attached code about the training is a little bit complicated. Do the authors have any plan to release the code or make it a component of PyTorch?

**Main Review:**

Originality: The task is new. The method combinates some well-known techniques and also proposes its own technique part. It also makes a clear statement about the differences from previous contributions.

Quality: The submission is technically sound.  Most claims are well supported (e.g., by theoretical analysis or experimental results). This is a complete piece of work. The authors are careful and honest about evaluating both the strengths and weaknesses of their work.

Clarity: The submission is clearly written and well organized.

Significance: The results are important and others would like to use the ideas and build on them.
The problem has lots of meanings for engineering implementation.
The experimental results are solid and strong.
The analysis and the experiments of the proposed VR-PGE are solid and convincing.

**Time Spent Reviewing:**

3 hours

---

> ### Author Response · Authors · 2021-08-10
> **Reply to Reviewer CTnr**
>
> Thanks for your recognition on our work.
>
> We will clean and release our code after acceptance.

---

> > ### Comment · Reviewer_CTnr · 2021-08-24
> > **Reviewer Response**
> >
> > After reading the comments from other reviewers and responses from authors, I reduce the score from 8 to 7. But it is still a good paper worth accepting.

---

### Official Review · Reviewer_X59J · 2021-07-15

**Rating:** 5
**Confidence:** 3

**Summary:**

Network sparsity has historically targeted only inference.  Recent explorations in training with sparsity fall short of their potential, though, by not fully exploiting sparsity in the forward and backward propagation steps.  The authors decompose training a channel-wise sparse network into two parts: training the weights themselves (as usual) and learning the sparse structure.  Rather than using a chain-rule based gradient step for the structure, a unique policy gradient estimator with reduced variance is used; it is composed of two forward passes, eliminating the need for the chain rule to be used to learn the sparse structure.  Experiments show competitive accuracy for saved parameters and FLOPs for a number of networks, as well as actual time saved during training with the proposed technique for VGG19 on CIFAR-10.

**Limitations And Societal Impact:**

There's no discussion of limitations in this work.  For example:
- Can the proposed technique be applied to transient values, such as activations or attention layers (which have no learnable parameters), to reduce training (and inference) costs?  In recent thrusts for many domains, including image classification, such approaches are proving to be very successful.
- Are there limitations on optimization schemes that will and will not work with the proposed technique?

I'm sure the authors are better able to identify the limitations of their work than I am.

**Main Review:**

# Originality

There's been lots of work in sparse training.  The authors present a nice survey in Section 2.2, but they've missed a number of works which do exploit sparsity in backpropagation [1-4].  It's likely that the recent works [3,4] could be trivially modified to support channel-pruning, rather than fine-grained sparsity.  What may be new here, and what the authors should clarify, is that the proposed approach is more theoretically grounded, without relying on heuristics to choose sparsity that isn't consistent between the forward and backwards passes.  Note: I'm not positive about this point; the burden remains on the authors to argue this convincingly!

I'm not familiar with any other work on reducing the variance of policy gradient estimation, though, so at least this contribution was new to me.

# Quality
Aside from the missing references that cast doubt on the claims of novelty, the experiments show that the technique does what it intends to do: variance of the gradients are reduced compared to the baseline PGE, accuracy is plausibly maintained as the number of nonzero parameters are reduced, and training time is reduced in practice.  In particular, reducing the training cost of ResNet50 on ImageNet1K by 1.6x is no small feat!  (In practice, this will be greatly reduced, of course.)

Remark 2 is potentially misleading.  It suggests that the authors of GrowEfficient used a dense backprop procedure, even for zero parameters.  Is this just for simplicity of implementation, though?  If so, it's no cause for penalization, as computing with zeros is a common way of testing an algorithm; exploiting the zeros often follows.

There's no discussion of hyperparameters p and a, how sensitive results are to their values, and suggestions on how to select them.  A step in the right direction would be to detail (in the main text) the settings used for the experimental results.

# Clarity
I found some parts of the manuscript confusing and hard to follow.
- Section 3 uses a fully-connected layer for simplicity (and it's understood that a convolution is a particular type of FC layer), but then Section 4 immediately switches to discussing in the context of convolutions.  Why the change?  Sticking with FC layers would help ease understanding.
- Line 122: "It is obvious that these methods cannot achieve significant speedup in training since they need to calculate the full gradient in backward propagation."  It's not obvious to me!  It sounds like the argument is that since the filters are not actually zero, neither are their gradients.  In the same sentence, though, it is said that "forward propagation could be sparsified if implemented carefully."  If these not-quite-zero values can be exploited for fprop, why couldn't the same happen in bprop?
- Similarly, the difference that gives true sparsity in the proposed method's backprop step, where prior efforts could not realize this sparsity, is not clear to me.  The authors suggest that since the output of the pruned channel is zero (Line 196), backprop can be sparse.  Is this not also the case in equation 1?  The term marked "forward" is zero for the pruned neuron, so the output should similarly be zero.

It'd be helpful if all equations were numbered for easy reference.

There were also a number of typos; another editing pass would be helpful to fix these (and, likely, others that I missed):
- Line 70: "… zero valued mask*s*"
- Line 74: "propagation*S*"
- Line 121: "filers" -> "filters"
- (Maybe not a typo, but a style thing) Figure 1's caption uses "3-rd," when "3rd" is more common, or, in this instance, "third" would also be appropriate.
- Algorithm 1, line 1: "p1" (is this supposed to be simply "p"?)
- Section 5.5 mentions Top-1 accuracies for MobileNetv2 on ImageNet, but I think these numbers (90%+) are for Top-5.  (Why switch between Top-1 and Top-5 for Figure 3's results and discussion?)

# Significance
Saving training time while still generating a network that maintains accuracy is an important topic.  This work is no magic bullet, though, because accuracy doesn't quite match the dense baseline accuracy, even for the more modest savings.  However, it does seem much more promising than the work to which it is compared, in both accuracy vs. parameters and achieved training time savings, so it may be a foundation on which future work improves.  However, there are some baselines missing both qualitative and quantitative comparisons, so the importance of this work may be reduced with the added context.

---

[1] "meProp: Sparsified Back Propagation for Accelerated Deep Learning with Reduced Overfitting," Sun et al.,  ICML 2017

[2] "Structurally Sparsified Backward Propagation for Faster Long Short-Term Memory Training," Zhu et al., https://arxiv.org/abs/1806.00512

[3] "Sparse Weight Activation Training," Raihan and Aamodt, NeurIPS 2020

[4] "Top-KAST: Top-K Always Sparse Training," Jayakumar et al., NeurIPS 2020


**Time Spent Reviewing:**

5

---

> ### Author Response · Authors · 2021-08-10
> **Reply to Reviewer X59J (2)**
>
> **Q8: The authors suggest that since the output of the pruned channel is zero (Line 196), backprop can be sparse. The term marked "forward" is zero for the pruned neuron, so the output should similarly be zero.**
>
> **A8:** It is a misunderstanding. In the equation in line 195, our point is: since $x_{out, c}$ is 0 regardless of what $F_c$ is, the gradient of $F_c$ is 0 and hence we do not need to compute it. Thus the backpropagation of weights is sparse.
>
> As for equation (1),**for pruned neuron, $m_3=0$ but the term marked "forward" is not 0**. The reason is that $x_{in}$ consists of both pruned ($x_{in, c}$) and unpruned output, so only part of $x_{in}$ is zero. This makes the item $w_{:,3}^T x_{in}$ in equation (1) nonzero, so the term marked as forward is not zero and the output is not zero. Therefore the backprop process of gradients is dense. This is the difference.
>
> **Q9: Typos.**
>
> **A9:** Thanks for your comments. However, the $\textbf{1}$ in $\rho\textbf{1}$ is not a typo. It is a vector with all values equalling to one. This is needed as $\textbf{s}$ is a vector and $\rho$ is a scalar. We miss a detailed caption for Figure 4, the third and fourth subgraph of Figure 4 is on VGG-19 on CIFAR-10. It is Top-1 accuracy.
>
> **Q10: It'd be helpful if all equations were numbered for easy reference.**
>
> **A10:** We will number all equations.
>
> **Q11: Doesn't quite match the dense baseline accuracy, even for the more modest savings.**
>
> **A11:** The searched model can match up with the dense model, if we relax the pruning ratio further. Actually, in this field, we usually evaluate the performance of our methods with others by comparing the accuracy at the same sparsity instead of keep the accuracy of the dense model. The intent of our work is to demonstrate that we can achieve much speed-up boost against prior works with even better accuracy, params and FLOPs.
>
> **Q12: However, there are some baselines missing both qualitative and quantitative comparisons, so the importance of this work may be reduced with the added context.**
>
> **A12:** GrowEfficient is the SOTA method on sparse training. To the best of our knowledge, no significant speedups in training are reported in the literature. The proprosed references [1-4] have been fully discussed in Q2. If you find some exising methods demonstrated with significant practical speedups (e.g., larger than 3) in training, please tell us and we are willing to compare with them.
>
>
> **Q13: Limitations.**
>
> **A13:** Actually we have discussed the limitations in appendix. As for the question one you raised, our method is not applicable to cases without structure parameters. As for the question two, we find Adam and SGD both works for the optimization of structure parameters, while Adam performs a litter bit better.

---

> ### Author Response · Authors · 2021-08-10
> **Reply to Reviewer X59J (1)**
>
> **Q1: Originality and novelty of our method.**
>
> **A1:**
> We notice that most of the reviewers have some concern about our statement that the proposed method is "the first one that can exploit the sparsity to significantly accelerate the training process in practice"(line 80). We think it is not a fundamental issue and it comes from our different understanding about "truly sparse", "significantly accelerate" and "in practice". We certainly acknowledge the contributions of the existing methods while we would like to clarify our understanding/definition about "truly sparse" here and we will also modify our statements in the final version accordingly. To ensure "significant speedup in practice", we think "truly sparse algorithm" should satisfy the following **three criteria**:
>
> **1) The computation in both forward and backward propagations should be completely sparse, i.e., the computational complexity should be significantly lower than that in standard training (Remark 1).** Especially the gradients to weights should be sparse rather than dense. This excludes algorithms like SWAT, SoftFilter.
>
> **2) During the whole training procedure, the training algorithm should always work on small sub-networks with the target sparsity, i.e., both the forward and backward propagations only need to  go through a small sub-network instead of the dense network (line 233).** The reason why we set this criterion is that both memory usage and computational cost of  such totally sparse training algorithm are bounded  from beginning to the end of training, leading to significant savings in both memory usage and computational cost and leaving broad space for exploring larger models.  This excludes dense-to-sparse training methods like PruneTrain (Lym, Sangkug, et al. 2019) and ClickTrain (Zhang, Chengming, et al., 2021), which consume huge memory and heavy computation in the early stage.
>
> **3) The training algorithm should be able to be implemented easily on the widely-used platforms on Pytorch or Tensorflow, since  most researchers today use these two platforms to train neural networks and explore new models (line 60).** Although many weight-level sparse training methods lead to speed-up in training theoretically, the real implementation of such algorithms involves huge efforts inaccessible for individual researchers or engineers. Therefore, developing easy-to-use sparse training algorithms on these two platforms is meaningful for broad audience. In contrast, our algorithm can be easily implemented on Pytorch and Tensorflow, and we will release our code on acception.
>
> We summerize previous methods into different groups:
>
> | Weight-level  |  |Channel-level| |
> | ----------- | ----------- | ----------- | ----------- |
> | |  Non-parametric        |  Parametric |  Parametric|
> |   |               |      Previous methods       |Our method |
> | Not applicable to widely-used platforms like PyTorch and Tensorflow, lack of support from sparse backward propagation for convolutional neural networks. | Cannot achieve significant speed-up constrained by calculating dense gradients of weights and dense-to-sparse training paradigm.       |Calculating gradients of structure parameters by dense backward propagation. | Calculating gradients of structure parameters by sparse forward propagation.|
> |Violates criterion 3.    |   Violates criteria 1,2.    |Violates criteria 1.      | The first to satisfy all three criteria. |
>
>
> To the best of our knowledge, our method is the first to satisfy the three criteria at the same time. Besides, as you metioned, compared with previous methods our method is more theoretically grounded and can achieve significant speedups in training.
>
>
> We hope the clarification above assist in resolving your concerns and better evaluating our paper.
>
>
> **Q2: Discussion on [1-4].**
>
> **A2:** The ideas of [1,2] are interesting, they aim at sparsify backpropagation. But it is unclear whether they can work well on channel level or can be applied to pruning.
>
> As you mentioned, [3,4] focus on weight-level speed-up and adopt the weight magnitude as the pruning metric. Existing result (Pruning Filters For Efficient Convnets) shows that directly extending the weight magnitude to the norm of filter to prune channel cannot achieve SOTA performance. Besides, [3] computes gradients in a dense way which precludes huge speed-up, that is 1/3 computation of forward and backward is dense, limiting the theoretical train-cost saving under 3. To summerize, [3] violates the criteria 1 and 3, and [4] violates the criterion 3. In contrast, our method can speed up the training process significantly in practice.
>
> **Q3: GrowEfficient requires dense backpropagation.**
>
> **A3:** The backpropagation for computing the  **gradients of weights** can be sparse, following the same logic as Section 4.2.1. The dense part of computation comes from the dense backpropagation to compute the **gradients of structure parameters**. That is, from equation (3), we find that the computation of gradients of structure parameters involves dense backpropagation process and partially sparse forward propagation process. It is not an issue of implementation but an issue of algorithm: if the gradients of $q_{c}$ with $s_{c}=0$ are not calculated to update $q_c$, GrowEfficient will collapse.  We present experimental results as follows:
>
> |  | Top-1 Acc(%) | FLOPs (%) |
> | ----------- | ----------- | ----------- |
> | GrowEfficient (update all structure parameters)| **81.8**        | 1.60 |
> | | **75.98**       | 0.99 |
> | GrowEfficient (update structure parameters with mask = $1$)|  10.0      |      3.69         |
> | |       10.0      |      1.38        |
>
> The reported result of GrowEfficient cannot be reproduced by only updating structure parameters with mask = $1$.
>
> Intuitively, we can understand it from a simple case. In Eqn.(6) of GrowEfficient (ICLR2021 version), suppose that some channel $c$ has a very very small score $s_c$ and the corresponding mask $q_c=0$. On one hand, if we do not calculate the gradient of $s_c$ and update it in backward propagation, then $s_c$ would stay small during the whole training process. This can make $q_c$ always be 0 and behave like a deterministic/dead mask, easily making the training failed. On the other hand, if the loss can change significantly by flipping $q_c$, we can expect that the gradient of $s_c$ is nonzero. Therefore, in fact, the gradient of $s_c$ can be nonzero even if $q_c=0$. Hence, if we calculate the gradient in backward propagation, the computation has to be dense. You may double-check with the authors with the update rule through email.
>
>
> **Q3: Suggestions on how to select hyperparameters.**
>
> **A3:** $\rho$ (remaining ratio) is determined by the budget (FLOPs, parameters) of the device in real applications. For a research paper, we just need to vary the value of $\rho$ to evaluate the performance at different sparsity. We choose the $\rho$ to obtain a model with comparable FLOPs and parameters. We will present detailed values of experimental results accordingly. As we specified in line 226, our algorithm works well by simply letting $\alpha=1/2$.
>
> **Q5: Fully-connected layer for simplicity in Section 3.**
>
> **A5:** We use fully-connected network instead of convolutional network just for simplifying the equations, that is we can simply the derivative of the convolution operator into that of scalar product. Actually, convolutional layer can be viewed as generalized fully-connected layer, by generalizing scalar product to matrix convolution. In this view, channel  pruning in convlutional network corresponds to the neuron (not weight) pruning in generalized fully-connected network. Therefore, to understand the case of convolutional network, you only need to replace the derivative of scalar product  with derivative of convolution, others are exactly the same.
>
> **Q6: About line 122.**
>
> **A6:** You can understand this from the training of lasso. Part of weights maybe zero during the training process, this can lead to sparse forward propagation. However, the gradient of all the weights (regardless of whether weights are 0 or not) need to be calculated, since the gradients of zero-valued weights can still be non-zero. So the backward propagation has to be dense. But for forward propagation, we do not need to calculate the products of the input with the zero-valued weights since they are 0. This is the key difference.
>
> **Q7: Why our method is sparse for gradient estimation of structure parameters and weights.**
>
> **A7:** Firstly we need to point out that the challenge of exploiting the sparsity lies in computing the gradient of the structure parameters. The main difference between our method and prior efforts is that we use forward propagation to compute this gradient while prior efforts use backward propagation.   To be precise:
>
>
> **1)** In our method, we use forward propagation to calculate the gradient of the structure parameters $\mathbf{s}$ and backward propagation to compute the gradient of the weights. (1) When we use forward propagation to compute the gradient of $\mathbf{s}$, since the sampled mask $\mathbf{m}$ is sparse, we do not need to compute the convolution associated with the pruned channel. Therefore, forward propagation can be sparsified. (2) Then for backward propagation to compute the gradient of the weights, you can high levelly  understand it as follows. If the mask $m_c$ of channel $c$ is zero, then disturbing weight $F_c$ would not change the network output ($F_c*m_c$ is always 0), i.e., the gradient of $F_c$ is $0$. Therefore, we do not need to go through the pruned channels in backward propagation. For the rigorous proof, please see Section 4.2.1.
>
> **2)** For the prior effort, they calculate the gradient of structure parameters through backpropagation (i.e., equation (1), it is a dense process) instead of forward propagation as we do. The reason of why it is dense is given in the answer of next question Q8.

---

> > ### Comment · Reviewer_X59J · 2021-08-25
> > **Missed some details of suggested reference**
> >
> > Hi, authors, and thank you for your in-depth response.  You've addressed some of my concerns, however I believe you may have missed some of the details of suggested reference [3] - in fact, I missed them when I suggested it!  Particularly, [3] *does* present a channel-pruning methodology, and achieves better accuracy and similar FLOPs reduction to your own method.  From Table 5 in [3] and Table 3 in your submission (along with the updated baseline produced under your environment from https://openreview.net/forum?id=JnAU9HkXr2&noteId=1yfXGW1_NrP ):
> >
> > |  Method  | Channel Sparsity | Baseline Top-1 | Pruned Top-1 | Training FLOPs Reduction (%) |
> > |:--------:|:----------------:|:--------------:|:------------:|:----------------------------:|
> > | SWAT [3] |              50% |           76.8 |         76.5 |                         47.6 |
> > | Proposed |            48.2% |           77.0 |         76.0 |                         53.2 |
> >
> > > 1) The computation in both forward and backward propagations should be completely sparse, i.e., the computational complexity should be significantly lower than that in standard training (Remark 1). Especially the gradients to weights should be sparse rather than dense. This excludes algorithms like SWAT, SoftFilter.
> >
> > > ...
> >
> > > To summerize, [3] obeys the criteria 1 and 3
> >
> > Can you clarify?  These statements are at odds with each other.
> >
> > > Besides, [3] computes gradients in a dense way which precludes huge speed-up, that is 1/3 computation of forward and backward is dense, limiting the theoretical train-cost saving under 3.
> >
> > Though the pruned weights are updated (the output of the weight gradient computation is dense), the weight gradient computation itself is sparse. This enables theoretical speedups larger than three, as seen in many of the results, such as Tables 2 and 3, which describe Training FLOPs reductions of around 90% in some cases.  The weight update computation is an elementwise addition operation, which costs significantly less than the convolution required to produce those weight gradients.
> >
> > So, it really seems that SWAT [3] is very competitive with your method.  From the limited results I've pasted here, it's not clear which is superior.
> >
> > >  Actually we have discussed the limitations in appendix.
> >
> > This is a minor issue, but this sort of content belongs in the main paper.  A good guideline is that if a section is required for submission (or strongly encouraged, as indicated by the required checklist), then it is more important than appendix content.

---

> > > ### Author Response · Authors · 2021-08-28
> > > **Reply to Reviewer X59J (3)**
> > >
> > > Thanks for your reply and actively discussing with us.
> > >
> > > **Q14: [3] does present a channel-pruning methodology.**
> > >
> > > **A14:** The idea of [3] to sparsify the weights and activations at the same time is interesting. **However,  the method [3] presents is not standard channel-level pruning. Its granularity of pruning is between the  standard channel pruning and weight pruning.** Thus, comparing channel pruning methods like us with [3] is unfair. The words 'CP: channel pruning' in the caption of  Table 5 of [3] actually do not refer to standard channel pruning. We can know this from the  evidence below:
> > >
> > > In fact, the authors stated in Section 3.3.1  of [3] that **"the importance of channels is considered independently, i.e., different filters can select different active channels"**. Figure 3 in [3] illustrates this difference between their method and standard channel pruning:
> > >
> > > (1) Figure 3 indicates that [3]  prunes the last $2\times 2$ matrix in the first row and the first $2\times 2$ matrix in the second row.
> > >
> > > (2) In standard channel pruning, we are only allowed to prune some entire rows instead of some matrices in one row.
> > >
> > > Mathematically, if we denote the weights of filters in one layer as $\{F_i\in \mathbb{R}^{k\times k\times C_{in}}: i = 1, \ldots, C_{out} \}$ with $C_{in}$ and $C_{out}$ being the numbers of input and output channels, respectively. Then, [3]  prunes some $k\times k$ matrices for each $F_i$ independently, while standard pruning methods are only allowed to prune some entire $F_i$ of these $C_{out}$ filters. This makes [3] has much more freedom in pruning than channel pruning methods. For example, if we have 128 input channles, i.e. $C_{in}=128$, then in pruning each $F_i$, theoretically, [3] has $2^{128}$ choices (i.e., 2 choices for each $k\times k$ matrix, totally $2^{128}$), while channel pruning methods only have 2 choice (i.e., prune or keep $F_i$).
> > >
> > > The reason of why [3] prunes in this special way may be that $L_1$ norm of $k\times k$ matrix could be a good metric to evaluate the importance of each of these $C_{in}$ matrices sized of $k\times k$ in $F_i$, while $L_1$ norm of whole $F_i$ is too rough for evaluating the importance of these $F_i$s.
> > >
> > > In summary, the granularity of pruning of [3] is between weight pruning and  channel pruning. The consequences are:
> > >
> > > (1) In both training and inference, the sparisity of [3] cannot be implemented by Pytorch and Tensorflow using the standard covolutional operator.
> > >
> > > (2) The pruned network obtained by [3] is not a standard convolutional network since we cannot remove some entire channel from the dense network.
> > >
> > > (3) It  is unfair for standard channel pruning methods to compare with [3] as [3] has much more freedom in pruning. A similar example is that weight pruning methods can always achieve higher accuracy than channel pruning methods at the same pruning rate, the reason is weight pruning methods can prune the network in a much finer granularity.
> > >
> > >
> > >
> > > **Q15: [3] achieves better accuracy and similar FLOPs reduction to your own method.**
> > >
> > > **A15:** As we point out in A14, the granularity of pruning of [3] is much finer than  channel pruning, such comparison  is actually unfair.  In addition, [3] cannot achieve training speed up on Tensorflow and PyTorch since it cannot be implemented by using their standard convolution operator.
> > >
> > > **Q16: These statements are at odds with each other.**
> > >
> > > **A16:** Thanks for pointing out this. Actually, the "obey" should be “violate” and we have fixed this typo now.
> > >
> > > **Q17:** 1/3 computation of forward and backward is dense.
> > >
> > > **A17:** We can divide the computation into 3 parts: forward propagation, gradient of the activation, gradient of the weights. For most weight-level methods, the gradient calculation of  the weights is dense, therefore, we say 1/3 of the computation is dense.
> > >
> > > Actually, for [3], you are correct. Since [3] is quite different from other methods as it sparsifies the activation as well, the gradient computation can thus be sparsified according to equation (3) in [3]. We made a mistake on this since we missed some details on [3]. Thanks for telling us  and we hope this won't produce negative effects on us as [3] is not an appropriate  baseline for us as we discussed.
> > >
> > > **Q18:  For [3] and this paper,  it's not clear which is superior.**
> > >
> > > **A18:** As we pointed out in A14, [3] prunes on a much finer granularity than channel pruning, which is not well supported by Tensorflow and PyTorch. In contrast, our method can achieve practical training speedups and  good accuracy at the same time. Therefore, our method is superior.
> > >
> > > **Q19:  Discussion of limitation belongs in the main paper.**
> > >
> > > **A19:** Thanks for your reminder. We will move it back to the main paper.
> > >
> > > Thanks again for your constructive comments on our work. If you feel this rebuttal resolves all your concerns, we appreciate it if you could raise your ratings. If you have any other concerns, please let us know and we are happy to discuss with you.

---

### Official Review · Reviewer_ozBF · 2021-07-16

**Rating:** 7
**Confidence:** 5

**Summary:**

Paper proposes a variational based pruning method where, with 2 structurally sparse forward passes and 1 sparse backward, the training can be significantly faster. Description is clear and theoretically sounds, equations are easy to follow, code is provided, results are encouraging. The paper might be a great benchmark and starting point for faster pruning-while-training with efficient inference.

**Main Review:**

References need more work. For example many papers do not have the venue(journal) where it was published.

# On sparse neural network training

Authors provide an overview of existing methods. However almost all structured pruning algorithms can be applied after few echoes, dense model obtained by removing channels, and then the rest of the training will continue with the small model. This approach will speed training as well. Are authors aware of such methods? For example [R1-R2]. These methods finish pruning with only 30% of training done, making the last 70% to be truly sparse.
Claiming to be the first in accelerating training via pruning might be challenged looking at previous work.

# Approach

The method theoretically sound. Equations are clear to be followed, Algorithm 1 explains the implementation clearly. Explanation and theory behind VR-PGE is interesting and clearly explained to be motivated by mean field theory.

K (in line ) controls the pruning ration of the layer. If this is true then we have 1 extra parameter to finetune.

# Results

Comparison shows better performance of the proposed approach over other pruning methods on CIFAR10/100. However, pruning network while training on CIFAR is not practical. Speeding up ImageNet training is more practical.

When training ResNet-50 with accelerated strategy it will be nice to see a comparison with training a smaller model from scratch. For example, training ResNet-50 with uniform channel reduction initialization (only 50%, 75% of initial channels in the model should be 1.6x faster). One more obvious baseline is to finish training earlier for ResNet50. For example doing only half of training epochs. Cosine (or monotonic) learning rate decay will remove the need of hyper parameter search.

When comparing results with pruning methods more other approaches should be added. For example, R3 has 76.4% with 50% FLOPs reduction. This method doesn't make pruning faster,

# Overall

The approach is interesting, clearly explained and theoretically sound. Reviewer is not too much familiar with variation methods for pruning and might miss some important references. The approach of pruning aware training that speeds up training is not novel, hovewer the method is novel.

Question if the pruning strategy is global (setting up K to be the neuron ration for the entire model, global pruning) or per-layer (setting K for every layer independently). In the former case cross-layer dependency might affect results and will need more hyper parameters. Code seems to indicate global pruning, just want to be sure.

Also additional baselines are required to see if the method outperform smaller model training from scratch in terms of training cost reduction.
For example ResNet-50 with uniform scaling, finishing training earlier (would be easy to target a particular 1.6x speed up, ImageNet preferably)

Finally the baseline accuracy of ResNet50 on ImageNet is somewhat low. With the setting of cosine annulling, 4 GPUs we should expect ~77.2 accuracy of the full model. Did authors train the Original model with their setting or took from the model zoo. The latest might make comparison not correct.

Providing code is great, however I found it a bit difficult to follow with too many arguments as input. Also for the final code release, would suggest to remove other than the proposed methods to make easy to follow.



[R1] S. A. Aketi, S. Roy, A. Raghunathan, and K. Roy. Gradual channel pruning while training using feature relevance scores for convolutional neural networks. IEEE Access, 8:171924– 171932, 2020.

[R2] Yang He, Yuhang Ding, Ping Liu, Linchao Zhu, Hanwang Zhang, and Yi Yang. Learning filter pruning criteria for deep convolutional neural networks acceleration. In Proceedings of the IEEE/CVF Conference on Computer Vision and Pattern Recognition, pages 2009–2018, 2020.

[R3] EagleEye: Fast Sub-net Evaluation for Efficient Neural Network Pruning, ECCV2020

**Time Spent Reviewing:**

4

---

> ### Author Response · Authors · 2021-08-10
> **Reply to Reviewer ozBF**
>
> **Q1: Missing publication information.**
>
> **A1:** We will add exact publication information accordingly.
>
> **Q2: Claiming to be the first truly sparse neural network training method.**
>
> **A2:**
> We notice that there is some concern about our statement that the proposed method is "the first one that can exploit the sparsity to significantly accelerate the training process in practice"(line 80). We think it is not a fundamental issue and it comes from our different understanding about "truly sparse", "significantly accelerate" and "in practice". We certainly acknowledge the contributions of the existing methods while we would like to clarify our understanding/definition about "truly sparse" here and we will also modify our statements in the final version accordingly. To ensure "significant speedup in practice", we think "truly sparse algorithm" should satisfy the following **three criteria**:
>
> **1)The computation in both forward and backward propagations should be completely sparse, i.e., the computational complexity should be significantly lower than that in standard training (Remark 1).** Especially the gradients to weights should be sparse rather than dense. This excludes algorithms like SWAT, SoftFilter.
>
> **2)During the whole training procedure, the training algorithm should always work on small sub-networks with the target sparsity, i.e., both the forward and backward propagations only need to  go through a small sub-network instead of the dense network (line 233).**
>     The reason why we set this criterion is that both memory usage and computational cost of  such totally sparse training algorithm are bounded  from beginning to the end of training, leading to significant savings in both memory usage and computational cost and leaving broad space for exploring larger models.  This excludes dense-to-sparse training methods like PruneTrain (Lym, Sangkug, et al. 2019) and ClickTrain (Zhang, Chengming, et al., 2021), which consume huge memory and heavy computation in the early stage.
>
> **3)The training algorithm should be able to be implemented easily on the widely-used platforms on Pytorch or Tensorflow, since  most researchers today use these two platforms to train neural networks and explore new models (line 60).** Although many weight-level sparse training methods lead to speed-up in training theoretically, the real implementation of such algorithms involves huge efforts inaccessible for individual researchers or engineers. Therefore, developing easy-to-use sparse training algorithms on these two platforms is meaningful for broad audience. In contrast, our algorithm can be easily implemented on Pytorch and Tensorflow, and we will release our code on acception.
>
> We summerize the previous methods into different groups:
>
> | Weight-level  |  |Channel-level| |
> | ----------- | ----------- | ----------- | ----------- |
> | |  Non-parametric        |  Parametric |  Parametric|
> |   |               |      Previous methods       |Our method |
> | Not applicable to widely-used platforms like PyTorch and Tensorflow, lack of support from sparse backward propagation for convolutional neural networks. | Cannot achieve significant speed-up constrained by calculating dense gradients of weights and dense-to-sparse training paradigm.       |Calculating gradients of structure parameters by dense backward propagation. | Calculating gradients of structure parameters by sparse forward propagation.|
> |Violates criterion 3.    |   Violates criteria 1,2.    |Violates criteria 1.      | The first to satisfy all three criteria. |
>
>
> To the best of our knowledge, our method is the first to satisfy the three criteria at the same time. We  notice that some existing methods can achieve some speedup in training by careful implementation. For example, a few channels have chances to be removed by weight-level algorithms when the weights in these channels are all pruned. For some dense update algorithms, one can also remove some channels if the corresponding weights are quite small for a long time. However, even with such careful implementations/ hand-crafted tricks, no significant speedups are reported in the literature. As shown in SNFS (Dettmers and Zettlemoyer [2020]), even on the simple classification task on CIFAR10, the speedup one can achieve can only be 1.07-1.32.  Therefore, compared with these methods, our method is more theoretically grounded and can achieve significant speedups in training without such heuristic tricks in implementation.
>
>
> We hope the clarification above assist in resolving your concerns in our claim.
>
> **Q3: Discussion on [R1-R2].**
>
> **A3:** R1-R2 can exploit the sparsity to speed up training, while the train-cost savings ratio is limited by 1/30\%=3.33, as the first 30\% epochs are dense. Moreover, according to Algorithm 1 of R2, R2 uses a pretrained dense model, therefore the overall train-cost saving is below 1.
>
> **Q4: Whether the pruning strategy is global.**
>
> **A4:** Yes the $K$ controls the number of channels globally. We don't need to tune remaining channels for different layers individually.
>
> **Q5: Comparing with training a dense model with smaller epochs.**
>
> **A5:** We present the result of training a dense model with 1/1.6 $\approx$ 62.5\% epochs.
>
> | Method  | Top-1 Acc(%) | FLOPs (%) | Train-cost Savings($\times$)|
> | ----------- | ----------- | ----------- | ----------- |
> | Dense(62.5\% epochs) |  70.6  |  100 | 1.6|
> | Ours   | **76.0**        | **46.8** | **1.6**|
>
> **Q6: Comparing with other pruning methods.**
>
> **A6:** As stated the computational time and memory cost cannot be reduced by these pruning methods. Our method focuses on boosting training speed while achieving comparable accuracy, params, FLOPs results as SOTA pruning methods.
>
> We present the comparison with MetaPruning, DMCP and EagleEye as follows:
>
> | Method  | Top-1 Acc(%) | FLOPs (%) | Train-cost Savings($\times$)|
> | ----------- | ----------- | ----------- | ----------- |
> | MetaPruning| 75.4        | 48.9 | -|
> | DMCP    |       76.2        |      53.7         | - |
> | EagleEye |  **76.4**  |  48.9 | -|
> | Ours   | 76.0        | **46.8** | **1.6**|
>
> We find that our method obtains 1.6$\times$ training speed-up while achieving results better than MetaPruning and comparable results against DMCP and EagleEye.
>
> **Q7: Accuracy of dense model.**
>
> **A7:** We take the dense model results directly from GrowEfficient and we will change the result into the baseline produced under our setting (77.01). Note that the results of compared methods are produced under the same setting (warmup, label smoothing, etc.), so there is no unfair comparison between our method and baselines.
>
> **Q8: Code.**
>
> **A8:** We will clean and release our code after acceptance.

---

> > ### Comment · Reviewer_ozBF · 2021-08-25
> > **response**
> >
> > I would like to thank the authors for detailed feedback.
> >
> > Multiple reviewers argue that claiming to be the first in "truly sparse" training is questionable. Multiple metrics are provided on what we can call "truly sparse" are provided but they are subjective and might be different if we ask different people. I would like to come to agreement on how to position the paper. Most reviewers questioned to be the "first" and over claiming the contribution because of that. Are authors willing to change their position in the revised paper?
> >
> > Resnet50 results on imageNet seem to be the most interesting. As pointed by the other reviewer model accuracy is too low, and authors provided an updated number of 77.01. This must be included into the paper, it will remove the claim that there is lossless speed up of 1.6x. Probably authors should find another compression factor that will give that. This seems like a potential major issue.
> >
> > If training is done with no training cost reduction, should we expect better accuracy as more updates are applied? Having such result for ResNet50 will improve empirical justification of results.
> >
> > What is the real speed up of training ResNet50 on ImageNet?
> >
> > Extra results are appreciated and seem to be reasonable.
> >
> > Please address my points related to ResNet50 so that I can keep by score of 7.

---

> > > ### Author Response · Authors · 2021-08-28
> > > **Reply to Reviewer ozBF (2)**
> > >
> > > Thanks for your positive comments and constructive advice on our work!
> > >
> > > **Q9: About our statement concerning ”first truly sparse”.**
> > >
> > > **A9:**
> > > Thanks for your constructive advice. As we stated earlier in the rebuttal, we will follow your advise and we are happy to modify our statements in our paper. Here is what we would like to solve the issue based on your feedbacks:
> > >
> > > **(1)** Replace our title by "Efficient Channel-level Sparse Neural Network Training".
> > >
> > > **(2)** Remove all the words, such as  "first", "truly sparse" and so on in the main text.
> > >
> > > **(3)** Remove all the controversial statements such as "existing methods cannot achieve significant speedups".
> > >
> > > **(4)** Claim in the beginning of this paper that we focus on channel-level pruning to avoid some confusions from weight-level methods.
> > >
> > > **(5)** Instead of giving new definition of 'truly sparse', we will only consider them as good properties of our method, which we will state as follows:
> > >
> > > **(5.1)** The computation in both forward and backward propagations of our method is  completely sparse, i.e., the computational complexity is significantly lower than that in standard training.
> > >
> > > **(5.2)** During the whole training procedure, our method  works on small sub-networks with the target sparsity, i.e., both the forward and backward propagations only need to  go through a small sub-network instead of the dense network.
> > >
> > > **(5.3)** Our method can be implemented easily on the widely-used platforms on Pytorch or Tensorflow.
> > >
> > > **We hope these modifications can solve the issue in our earlier statement. If you have any other suggestions, please let us know and we are happy to  change according to your suggestions.**
> > >
> > > **Q10: Accuracy of dense model.**
> > >
> > > **A10:** We will certainly include the 77.01 result into the paper. Besides, we notice that we didn't claim that there is lossless speed up of 1.6x. And we think maybe our old baseline misleads you and we are sorry about that.
> > >
> > > The results of  prune after training methods, e.g., EagleEye, imply that to achieve lossless compression from standard ResNet50, we need to keep a large model, making the speedup quite limited.
> > >
> > > Fortunately, we find that another path to achieve training speedup without accuracy drop is to start from a model wider than ResNet50 enabling us to explore in a larger space than standard ResNet50. We work on wider ResNet50, i.e., ResNet-50-1.15, and keep the remaining channel ratio the same with the original ResNet-50 model. The results are presented in the following table. The accuracy we achieved is 76.8\%, close to the baseline.
> > >
> > >
> > > | Model | Method | Top-1 Acc(%) | Inference FLOPs(%) | Train-cost Savings($\times$) |
> > > | ----------- | ----------- | ----------- | ----------- | ----------- |
> > > | ResNet-50 | dense | 77.0 | 100 |  1.00 |
> > > | ResNet-50-1.15 | Ours | 76.8 | **56.2** |  **1.34** |
> > > | ResNet-50-1.25 | Ours | **77.3** | 75.02  |  1.00 |
> > >
> > > **Q11:  Training done with no training cost reduction.**
> > >
> > > **A11:**
> > >
> > > **(1)** We find that training with more steps with the same training cost leads to a model with 76.7\% Top-1 accuracy, slightly lower than the baseline. The reason may be the low convergence rate of stochastic optimization methods.
> > >
> > > **(2)** We tried another method as we do in A10. To be precise, we work on ResNet-50-1.25 and in this case the training cost is equal to that of the standard ResNet-50. The result is given in the table above. It shows that we can achieve higher accuracy than ResNet-50.
> > >
> > > **Q12: Real speed up of training ResNet50 on ImageNet.**
> > >
> > > **A12:** We present the result in the table below. It shows that we can achieve more speedup when the prune rate increases.
> > >
> > > | Model | Method | Top-1 Acc(%) | Params(%)  | Inference FLOPs(%) | Train-cost Savings($\times$) | Train-computational Time Savings ($\times$)|
> > > | ----------- | ----------- | ----------- | ----------- | ----------- | ----------- | ----------- |
> > > | ResNet-50 | GrowEfficient | 75.2 |61.2 | 50.3       | 1.10 | 1.075 |
> > > |  | GrowEfficient | - |11.3 | 10.8       | 1.28 | 1.16 |
> > > |  | Ours | **76.0** |**48.2** | **46.8**       | **1.60** | **1.33** |
> > > |  | Ours | **69.3** |**10.8** | **10.1**       | **7.36** | **3.2** |
> > >
> > > "-" means that GrowEfficient collapses at such high pruning rate, 11.3 remaining parameters. The reason may come from the inaccurate gradient estimation by straight through approximator. We find that our method beats GrowEfficient steadily both in terms of train-cost savings and train-computational time savings. The gap between our method and GrowEfficient becomes larger as the pruning rate goes higher.
> > >
> > > Thanks again for your constructive comments on our work. We hope this rebuttal resolves all your concerns. If you have any other questions, please let us know and we are happy to discuss with you.

---

### Official Review · Reviewer_T3oD · 2021-07-17

**Rating:** 4
**Confidence:** 4

**Summary:**

The authors propose a channel-level sparse training algorithm that can effectively utilize the channel sparsity to accelerate both forward and backward propagation process. They divide the sparse training process into weight updating step and mask updating step. And solve the problem separately. More specifically, they develop a variance-reduced policy gradient estimator to update the trainable pruning mask without the need for backpropagation, and hence a sparse computation on both forward and backward propagation.

**Limitations And Societal Impact:**

Yes

**Main Review:**

Strength:
This paper shows an analysis that why previous parametric works are not able to utilize the sparsity to accelerate the training.
The authors evaluate their method on several representative DNNs and Datasets.
It is great that the authors also show actual training computational time rather than just using FLOPs to indicate the acceleration.
Their results seem decent when considering both accuracy and compression ratio.

Weakness:
In the paper, the authors claim that they are the first to achieve training acceleration in practice. However, that is not correct or not rigorous. The existing work PruneTrain [1], and ClickTrain [2] can also achieve training acceleration. The author should also cite those papers and compare with them.

For the result comparison part, the accuracy of the original (dense) models is too low. For example, the accuracy of the original model for ResNet50 on ImageNet is 76.1%. I understand this is a widely used official model. But that model is too old, and it was probably obtained by some legacy training recipe. Your model is trained with warmup, label smoothing, and maybe cosine lr scheduler, which may significantly boost the model accuracy. To avoid misleading the audience, it is more reasonable to show the dense model accuracy obtained by using the same training settings as yours.
For a similar concern, the reference pruning works that you choose are also too old, which cannot represent the SOTA results (e.g., MetaPruning [3], DMCP [4]).

Many references are missing publication information.

You mentioned the reference work NetAdapt in the caption of Table III, but you didn’t show the results.

The total training epochs and learning rate scheduler are missing. This may make your results less convincing.

It is good to see the accuracy and training speed if you directly train a small dense model with exactly the same model structure, layer width, and FLOPs as the remaining part (dense) of your sparse model.

[1] PruneTrain: Fast Neural Network Training by Dynamic Sparse Model Reconfiguration, SC’19
[2] ClickTrain: Efficient and Accurate End-to-End Deep Learning Training via Fine-Grained Architecture-Preserving Pruning, ICS’21
[3] MetaPruning: Meta Learning for Automatic Neural Network Channel Pruning, ICCV’19
[4] DMCP: Differentiable Markov Channel Pruning for Neural Networks, CVPR’20


**Time Spent Reviewing:**

4 hours

---

> ### Author Response · Authors · 2021-08-10
> **Reply to Reviewer T3oD**
>
> **Q1: First truly sparse neural network training method.**
>
> **A1:**
> We notice there is some concern about our statement that the proposed method is "the first one that can exploit the sparsity to significantly accelerate the training process in practice"(line 80). We think it is not a fundamental issue and it comes from our different understanding about "truly sparse", "significantly accelerate" and "in practice". We certainly acknowledge the contributions of the existing methods while we would like to clarify our understanding/definition about "truly sparse" here and we will also modify our statements in the final version accordingly if accepted. To ensure "significant speedup in practice", we think "truly sparse algorithm" should satisfy the following **three criteria**:
>
> **1)The computation in both forward and backward propagations should be completely sparse, i.e., the computational complexity should be significantly lower than that in standard training (Remark 1).** Especially the gradients to weights should be sparse rather than dense. This excludes algorithms like SWAT, SoftFilter.
>
> **2)During the whole training procedure, the training algorithm should always work on small sub-networks with the target sparsity, i.e., both the forward and backward propagations only need to  go through a small sub-network instead of the dense network (line 233).** The reason why we set this criterion is that both memory usage and computational cost of  such totally sparse training algorithm are bounded  from beginning to the end of training, leading to significant savings in both memory usage and computational cost and leaving broad space for exploring larger models.  This excludes dense-to-sparse training methods like PruneTrain (Lym, Sangkug, et al. 2019) and ClickTrain (Zhang, Chengming, et al., 2021), which consume huge memory and heavy computation in the early stage.
>
> **3)The training algorithm should be able to be implemented easily on the widely-used platforms on Pytorch or Tensorflow, since  most researchers today use these two platforms to train neural networks and explore new models (line 60).** Although many weight-level sparse training methods lead to speed-up in training theoretically, the real implementation of such algorithms involves huge efforts inaccessible for individual researchers or engineers. Therefore, developing easy-to-use sparse training algorithms on these two platforms is meaningful for broad audience. In contrast, our algorithm can be easily implemented on Pytorch and Tensorflow, and we will release our code on acceptance.
>
> We summerize the previous methods into different groups:
>
> | Weight-level  |  |Channel-level| |
> | ----------- | ----------- | ----------- | ----------- |
> | |  Non-parametric        |  Parametric |  Parametric|
> |   |               |      Previous methods       |Our method |
> | Not applicable to widely-used platforms like PyTorch and Tensorflow, lack of support from sparse backward propagation for convolutional neural networks. | Cannot achieve significant speed-up constrained by calculating dense gradients of weights and dense-to-sparse training paradigm.       |Calculating gradients of structure parameters by dense backward propagation. | Calculating gradients of structure parameters by sparse forward propagation.|
> |Violates criterion 3.    |  Violates criteria 1,2.    |Violates criteria 1.      | The first to satisfy all three criteria. |
>
>
> To the best of our knowledge, our method is the first to satisfy the three criteria at the same time. PruneTrain (Lym, Sangkug, et al. 2019) and ClickTrain (Zhang, Chengming, et al., 2021) work with a large model at the begining epochs and consume huge memory and heavy computation in the early stage. The results (Table 1 of PruneTrian and Fig. 12 of ClickTrain) show that the speedup ClickTrain can achieve is close to 1 and PruneTrain cannot preserve the accuracy well although it can achieve some speedup. In contrast, our method works on small sub-networks with the target sparsity from beginning to end of training. We will cite and discuss with those papers.
>
>
> We hope the clarification above assist in resolving your concerns and better evaluating our paper.
>
> **Q2: Accuracy of dense ResNet-50 model.**
>
> **A2:** We take the dense model results directly from GrowEfficient and we will change the result into the baseline produced under our setting (77.01). Note that the results of compared methods are produced under the same setting (warmup, label smoothing, etc.), so there is no unfair comparison between our method and baselines.
>
> **Q3: Comparison with SOTA results.**
>
> **A3:**
> The main contribution and novelty of our paper is to develop a sparse training algorithm which can achieve significant savings in time cost while achieving comparable accuracy and FLOPs of the final sparse model. We present the comparison with MetaPruning and DMCP as follows.
>
> | Method  | Top-1 Acc(%) | FLOPs (%) | Train-cost Savings($\times$)|
> | ----------- | ----------- | ----------- | ----------- |
> | MetaPruning| 75.4        | 48.9 | -|
> | DMCP    |       **76.2**        |      53.7         | - |
> | Ours   | 76.0        | **46.8** | **1.6**|
>
> We find that our method obtains 1.6$\times$ training speed-up while achieving accuracy and FLOPs better than MetaPruning and comparable results against DMCP. We would like to point our that both of these two methods cannot reduce the training time significantly, which demonstrates the superority of our method.
>
> **Q4: Missing publication information.**
>
> **A4:** We will add exact publication information accordingly.
>
> **Q5: Errata of NetAdapt.**
>
> **A5:** We will fix this errata.
>
> **Q6: Total training epochs and learning rate scheduler.**
>
> **A6:** Total training epochs follow the same practice with GrowEfficient. Learning rate scheduler is cosine.
>
> **Q7: Directly train a small dense model.**
>
> **A7:** We present the result of training a small dense model directly on VGG-19 on CIFAR-10.
>
> | Method  | Top-1 Acc(%) | FLOPs (%) | Train-cost Savings($\times$)|
> | ----------- | ----------- | ----------- | ----------- |
> | Ours| **93.46**        | 28.57 | 2.64|
> | Direct Training    | 92.6  | 28.57 | 3.50|
> | Ours| **93.11**      | 19.33 | 3.89|
> | Direct Training    | 92.47  | 19.33 | 5.17|
> | Ours| **92.23**      | 10.27 | 7.30|
> | Direct Training    | 91.2  | 10.27 | 9.74|
>
> From the table we find that direct training of the final searched small sparse model leads to slightly lower Top-1 Accuracy. Besides the train-cost saving of direct training is higher because our method needs two forward propagations. This validates the effectiveness of our method in finding valuable structure for pruning and further validates the lottery ticket hypothesis.

---

> > ### Comment · Reviewer_T3oD · 2021-08-25
> > **Reviewer Response**
> >
> > Thanks for your efforts in answering the questions. But I still have problems with the work:
> >
> > For the Q1. I think it is a fundamental issue. Just as you said, this is caused by different understanding, but this will also happen to other readers. Since there are other works that can achieve training acceleration, I think it is inappropriate and misleading to claim to be the “first” to exploit sparsity in training, even though with your self-defined “significantly accelerate” or “in practice”.
> >
> > For your answer to Q7, do you have ImageNet result? If not, can you also show the comparison between small dense results and your results under similar Training-cost on CIFAR? A ResNet model (e.g., WideResNet-28) should be better than VGG19 since VGG are barely used anymore.
> >
> > Since (significant) training acceleration is a major contribution claimed in the paper, I still don’t understand why you don’t compare with the baseline works that can achieve training acceleration (e.g., PruneTrian). In your rebuttal, you said those dese-to-sparse training methods are excluded since they consume huge memory and heavy computation in the early stage. If that's the case, I find it confusing why you compared to pruning works (e.g. L1-Pruning, Provable), which don't focus on training, let alone training acceleration.

---

> > > ### Author Response · Authors · 2021-08-28
> > > **Reply to Reviewer T3oD(2)**
> > >
> > > Thanks for your reply and constructive comments.
> > >
> > > **Q8: About our statement concerning ”first truly sparse”.**
> > >
> > > **A8:** Thanks for your constructive advice. As we stated earlier in the rebuttal, we will follow your advise and we are happy to modify our statements in our paper. Here is what we would like to solve the issue based on your feedbacks:
> > >
> > > **(1)** Replace our title by "Efficient Channel-level Sparse Neural Network Training".
> > >
> > > **(2)** Remove all the words, such as  "first", "truly sparse" and so on in the main text.
> > >
> > > **(3)** Remove all the controversial statements such as "existing methods cannot achieve significant speedups".
> > >
> > > **(4)** Claim in the beginning of this paper that we focus on channel-level pruning to avoid some confusions from weight-level methods.
> > >
> > > **(5)** Instead of giving new definition of 'truly sparse', we will only consider them as good properties of our method, which we will state as follows:
> > >
> > > **(5.1)** The computation in both forward and backward propagations of our method is  completely sparse, i.e., the computational complexity is significantly lower than that in standard training.
> > >
> > > **(5.2)** During the whole training procedure, our method  works on small sub-networks with the target sparsity, i.e., both the forward and backward propagations only need to  go through a small sub-network instead of the dense network.
> > >
> > > **(5.3)** Our method can be implemented easily on the widely-used platforms on Pytorch or Tensorflow.
> > >
> > > **We hope these modifications can solve the issue in our earlier statement. If you have any other suggestions, please let us know and we are happy to  change according to your suggestions.**
> > >
> > > **Q11: Directly train a small dense model on ImageNet.**
> > >
> > > **A11:** We present the result of training a small dense model directly on ResNet-50 on ImageNet.
> > >
> > > | Method  | Top-1 Acc(%) | Inference FLOPs (%) | Train-cost Savings($\times$)|
> > > | ----------- | ----------- | ----------- | ----------- |
> > > | Ours| **76.0**        | 46.8 | 1.60|
> > > | Direct Training    | 74.8  | 46.8 | 2.14 |
> > > | Ours| **73.5**      | 24.7 | 3.02|
> > > | Direct Training    | 72.0  | 24.7 | 4.05|
> > >
> > > From the table we obtain consistent conclusion between CIFAR-10 and ImageNet: direct training of the final searched small sparse model leads to slightly lower Top-1 Accuracy. Besides the train-cost saving of direct training is higher because our method needs two forward propagations. This validates the effectiveness of our method in finding valuable structure for pruning and further validates the lottery ticket hypothesis.
> > >
> > > **Q12: Compare with baseline PruneTrain.**
> > >
> > > **A12:** Thanks for pointing out this paper. We didn't notice PruneTrain before rebuttal, since it is published in International Conference for High Performance Computing, Networking, Storage, and Analysis, which is not a machine learning conference. We are happy to include it as a baseline.
> > >
> > > The baselines L1-Pruning, Provable mainly focus on final accuracy and high computational cost (e.g., start from a pre-trained model) is usually tolerable in them. We include them in the experiments since we want to see the accuracy one can achieve in this setting.
> > >
> > > Below, we give some comparison results between our method and PruneTrain.
> > >
> > > (1) The table below present the results on  CIFAR-10 with ResNet-32 .
> > >
> > > | Inference FLOPs | 36%| 22% | 7%| 2%|
> > > | ----------- | ----------- | ----------- |  ----------- | ----------- |
> > > | PruneTrain(Acc) | 92.78 | 91.18 | 85.08 | 74.18 |
> > > | Ours(Acc) | **94.27** | **93.83** | **91.83**     | **87.21** |
> > >
> > > | Inference FLOPs | 36%| 22% | 7%| 2%|
> > > | ----------- | ----------- | ----------- |  ----------- | ----------- |
> > > | PruneTrain(Train-cost Savings) | 1.91$\times$ | 2.95$\times$ | 7.03$\times$ | 9.78$\times$ |
> > > | Ours(Train-cost Savings) | **2.19$\times$** | **3.31$\times$** | **10.14$\times$**     | **35.73$\times$** |
> > >
> > > The two tables above show that the accuracy achieved by PruneTrain decreases rapidly as the pruning rate (resp. inference flop) increases (resp. decreases). The reason may be that as [3] is a  dense-to-sparse method,  the dense stage is computational expensive. To achieve significant overall training speed up, PruneTrain needs to be aggressive in the early dense stage of training to reduce the model size quickly. This would mistakenly discard some important weights in the early stage, finally leading to larger accuracy drop.
> > >
> > > The results above also demonstrate that our method can achieve better performance than PruneTrain in both accuracy and training-cost saving. The gap between them becomes larger when the pruning rate increases.
> > >
> > > (2)Table below presents the results on ImageNet.
> > >
> > > | Model | Method | Top-1 Acc(%) | Inference FLOPs(%) | Train-cost Savings($\times$) |
> > > | ----------- | ----------- | ----------- |  ----------- | ----------- |
> > > | ResNet-50 | PruneTrain | 75.08 | 36.00       | 1.5 |
> > > |  | PruneTrain | 60.3 | 11.3      | 3.6 |
> > > |  | Ours | **76.0** | **34.22**       | **1.6** |
> > > |  | Ours | **69.3** | **10.1**       | **7.36** |
> > >
> > > It shows that our method can achieve better performance than PruneTrain both in accuracy and training-cost savings. The reason could be similar as we discussed above.
> > >
> > > Thanks again for your constructive comments. We hope this rebuttal resolves all your concerns. If you have any other questions, please let us know and we are happy to discuss with you.

---

### Author Response · Authors · 2021-08-10
**To All Reviewers(1)**

We notice that most of the reviewers have some concerns about our title "Truly Sparse Neural Network Training" and our statement that the proposed method is "the first one that can exploit the sparsity to significantly accelerate the training process in practice"(line 80). We think it is not a fundamental issue and it comes from our different understanding about "truly sparse", "significantly accelerate" and "in practice". We certainly acknowledge the contributions of the existing methods while we would like to clarify our understanding/definition about "truly sparse" here and we will also modify our statements in the final version accordingly if accepted. To ensure "significant speedup in practice", we think "truly sparse algorithm" should satisfy the following **three criteria**:

**1) The computation in both forward and backward propagations should be completely sparse, i.e., the computational complexity should be significantly lower than that in standard training (Remark 1).** Especially the gradients to weights should be sparse rather than dense. This excludes algorithms like SWAT, SoftFilter.

**2) During the whole training procedure, the training algorithm should always work on small sub-networks with the target sparsity, i.e., both the forward and backward propagations only need to  go through a small sub-network instead of the dense network (line 233).**
The reason why we set this criterion is that both memory usage and computational cost of  such totally sparse training algorithm are bounded  from beginning to the end of training, leading to significant savings in both memory usage and computational cost and leaving broad space for exploring larger models.  This excludes dense-to-sparse training methods like PruneTrain (Lym, Sangkug, et al. 2019) and ClickTrain (Zhang, Chengming, et al., 2021), which consume huge memory and heavy computation in the early stage.

**3) The training algorithm should be able to be implemented easily on the widely-used platforms on Pytorch or Tensorflow, since  most researchers today use these two platforms to train neural networks and explore new models (line 60).** Although many weight-level sparse training methods lead to speed-up in training theoretically, the real implementation of such algorithms involves huge efforts inaccessible for individual researchers or engineers. Therefore, developing easy-to-use sparse training algorithms on these two platforms is meaningful for broad audience. In contrast, our algorithm can be easily implemented on Pytorch and Tensorflow, and we will release our code on acception.

We summerize the previous methods into different groups:

| Weight-level  |  |Channel-level| |
| ----------- | ----------- | ----------- | ----------- |
| |  Non-parametric        |  Parametric |  Parametric|
|   |               |      Previous methods       |Our method |
| Not applicable to widely-used platforms like PyTorch and Tensorflow, lack of support from sparse backward propagation for convolutional neural networks. | Cannot achieve significant speed-up constrained by calculating dense gradients of weights and dense-to-sparse training paradigm.       |Calculating gradients of structure parameters by dense backward propagation. | Calculating gradients of structure parameters by sparse forward propagation.|
|Violates criterion 3.    |   Violates criteria 1,2.    |Violates criteria 1.      | The first to satisfy all three criteria. |

To the best of our knowledge, our method is the first to satisfy the three criteria at the same time. We  notice that some existing methods can achieve some speedup in training by careful implementation. For example, a few channels have chances to be removed by weight-level algorithms when the weights in these channels are all pruned. For some dense update algorithms, one can also remove some channels if the corresponding weights are quite small for a long time. However, even with such careful implementations/ hand-crafted tricks, no significant speedups are reported in the literature. As shown in SNFS (Dettmers and Zettlemoyer [2020]), even on the simple classification task on CIFAR10, the speedup one can achieve can only be 1.07-1.32.  Therefore, compared with these methods, our method is more theoretically grounded and can achieve significant speedups in training without such heuristic tricks in implementation.


We hope the clarification above assist in resolving your concerns  and better evaluating our paper.

---

### Author Response · Authors · 2021-08-19
**To All Reviewers (2)**

Second Reply:

We notice that some reviewers (e.g. Reviewer CRsd) are concerned about **the rigority of our definition of "truly sparse" in rebuttal** since our three criteria are not sufficient and necessary condition to obtain speed-up:

--We set these criteria because they can **naturally induce significant speed-up**, while indeed they are not sufficient and necessary condition to obtain speed-up. These three criteria show the appealing features/advantages of our method. Actually we know that there are lots of difficulties (e.g., even giving a rigorous definition for ‘sparse’ is not easy. Less than 1%, 5% or …nonzero values?) in proposing a definition for sparse training as rigorous as behaving like a sufficient and necessary condition. Therefore, we do not intend to propose a new rigorous definition for the community and our main contribution is the efficient algorithm but not the definition. To solve it, we will change our statement in the final version to indicate that these are the distinctions/advantages of our method and we do not intend to propose a new rigorous definition for the community.


We also notice that Reviewer CRsd is concerned about whether weight-level sparse training for convolutional neural networks is already fully supported on PyTorch or Tensorflow. **We can obtain the answer by simple reasoning: if the weight-level sparse training is well-supported by PyTorch or Tensorflow for convolutional networks, then there is no need to develop channel-level sparse training method since weight-level can be more efficient on reducing the model size and computational cost. However, this is obviously not the fact as lots of researchers are still focusing on channel-level sparse training.** The reference[9] raised by reviewer CRsd can only support the backward propagation of sparse matrix multiplication but not general sparse convolution. For details you can check A16 to reviewer CRsd.

We hope the clarification above assist in resolving your concerns  and better evaluating our paper.

---

### Author Response · Authors · 2021-08-28
**To All Reviewers: We will modify our statement concerning "first truly sparse" in the paper.**

Thanks for your constructive advice. As we stated earlier in the rebuttal, we will follow your advise and we are happy to modify our statements in our paper. Here is what we would like to solve the issue based on your feedbacks:

**(1)** Replace our title by "Efficient Channel-level Sparse Neural Network Training".

**(2)** Remove all the words, such as  "first", "truly sparse" and so on in the main text.

**(3)** Remove all the controversial statements such as "existing methods cannot achieve significant speedups".

**(4)** Claim in the beginning of this paper that we focus on channel-level pruning to avoid some confusions from weight-level methods.

**(5)** Instead of giving new definition of 'truly sparse', we will only consider them as good properties of our method, which we will state as follows:

**(5.1)** The computation in both forward and backward propagations of our method is  completely sparse, i.e., the computational complexity is significantly lower than that in standard training.

**(5.2)** During the whole training procedure, our method  works on small sub-networks with the target sparsity, i.e., both the forward and backward propagations only need to  go through a small sub-network instead of the dense network.

**(5.3)** Our method can be implemented easily on the widely-used platforms on Pytorch or Tensorflow.

**We hope these modifications can solve the issue in our earlier statement. If you have any other suggestions, please let us know and we are happy to  change according to your suggestions.**

---

### Author Response · Authors · 2021-09-01
**To AC(3)**

Dear ACs,

Thanks for your hard work in reviewing our paper. We would like to report the current status to you due to the approaching deadline of discussion.

We address the main concerns of negative reviewers as follows:

**1**. Reviewer CRsd and X59J are concerned about the novelty of this paper. They claim that some existing work can solve the channel-level sparse training problem effectively. Actually this mainly comes from **their serious misunderstandings on existing works**:

**(1)** Reviewer CRsd claims that "Sparse GPU Kernels for Deep Learning" can also support sparse backward propagation of convolutional networks. By looking into this work, we find the authors stated clearly that they only support standard sparse matrix multiplication involving $1\times1$ sparse convolution but not general sparse convolution.

**(2)** Reviewer X59J claims that SWAT is able to prune networks at channel level. Actually, SWAT prunes at the granularity finer than channel. To be precise, for filter $F\in \mathbb{R}^{k\times k \times C_{in}}$, SWAT can prune $k\times k$ matrices individually while standard pruning methods can only prune the entire $F$. The biggest consequence is that the pruned network obtained by SWAT is not a standard convolutional network, which cannot be implemented by standard convolution operator. Moreover, it's also unfair to compare channel-level sparse training methods with SWAT due to their different granularity of pruning.

**2**. Reviewer T3oD advised us to compare with PruneTrain. We present the results in the rebuttal and show high superiority of our method on both small-scale and large-scale datasets CIFAR-10 and ImageNet-1K.

Considering there is no new feedback from reviewers after we point out **their serious misunderstandings**, we believe all concerns of them, especially those holding negative ratings, have been effectively solved by our feedback. We very much appreciate it if you could look into it ( **especially the serious misunderstandings of Reviewer CRsd and X59J** ) and also wish our paper can be judged correctly and objectively in terms of its contributions.

Sincerely,

Authors

---

### Decision · Program_Chairs · 2021-09-27

**Decision:**

Accept (Poster)

**Comment:**

This paper proposes several effective schemes to ensuring the sparsity of a deep network being trained during backward and forward propagation. The reviewers recommendations are somewhat polarized and there are some serious disagreements between some reviewer and the authors regarding fair comparison with many existing sparsity promoting techniques. Despite some concerns, the proposed methods are relatively novel and effective and have been justified with adequate analysis and empirical evidences. The AC believes the quality of the paper is above the acceptance threshold.